# SAIL-RL: Guiding MLLMs in When and How to Think via Dual-Reward RL Tuning

## Abstract

We introduce SAIL-RL, a reinforcement learning (RL) post-training framework that enhances the reasoning capabilities of multimodal large language models (MLLMs) by teaching them when and how to think. Existing approaches are limited by outcome-only supervision, which rewards correct answers without ensuring sound reasoning, and by uniform thinking strategies, which often lead to overthinking on simple tasks and underthinking on complex ones. SAIL-RL addresses these challenges with a dual reward system: the Thinking Reward, which evaluates reasoning quality through factual grounding, logical coherence, and answer consistency, and the Judging Reward, which adaptively determines whether deep reasoning or direct answering is appropriate. Experiments on the state-of-the-art SAIL-VL2 show that SAIL-RL improves reasoning and multimodal understanding benchmarks at both 4B and 8B scales, achieving competitive performance against commercial closed-source models such as GPT-4o, and substantially reduces hallucinations, establishing it as a principled framework for building more reliable and adaptive MLLMs.

## 1 Introduction

Multimodal Large Language Models (MLLMs) (Comanici et al., 2025; Hurst et al., 2024; Bai et al., 2025; Liu et al., 2023; Chen et al., 2024) are continuously advancing, evolving from elementary visual description toward complex reasoning and comprehensive visual understanding. A major driver of this progression is the development of training paradigms and optimization strategies that determine how MLLMs acquire such capabilities from large-scale multimodal data. Supervised fine-tuning (SFT) with teacher-forcing laid the foundation for pre-training by aligning models with multimodal corpora. Building on this foundation, recent studies (Team et al., 2025a; Yang et al., 2025b) have increasingly emphasized post-training, where hybrid frameworks integrate SFT with reinforcement learning (RL). RL post-training has undergone a paradigm shift: early approaches (Ouyang et al., 2022; Rafailov et al., 2023; Ethayarajh et al., 2023) primarily focused on aligning models with human preferences, whereas recent methods (Yang et al., 2025b; Chen et al., 2025a; Team et al., 2025b) emphasize step-by-step thinking and iterative self-improvement. This transition enables models to refine their cognitive processes more autonomously, making RL post-training a particularly promising pathway for further advancing MLLM capabilities.

RL in MLLMs (Deng et al., 2025; Chen et al., 2025a; Team et al., 2025b) commonly follows the paradigm of *"thinking before speaking."* Guided by a special token \think, the model first generates a structured reasoning trace before producing the final answer. Leveraging long reasoning chains as an internal knowledge source allows the model to extract salient cues that improve answer accuracy and strengthen overall capability. Nevertheless, despite these advances, current RL post-training methods still face several fundamental challenges:

Answers without sound reasoning: Conventional methods rely on *outcome-only supervision*, where rewards are determined by the correctness of the final answer while the quality of reasoning is disregarded. This paradigm introduces two critical issues: first, as the intuition *"think well to answer right"* suggests, incoherent or redundant reasoning traces hinder the model from extracting useful cues, leading to inaccurate answers and exacerbating hallucinations. As shown in Figure 2, conventional MLLMs (Team et al., 2025b) can produce correct answers despite factual errors in reasoning, highlighting how outcome-only rewards compromise robustness and trustworthiness.

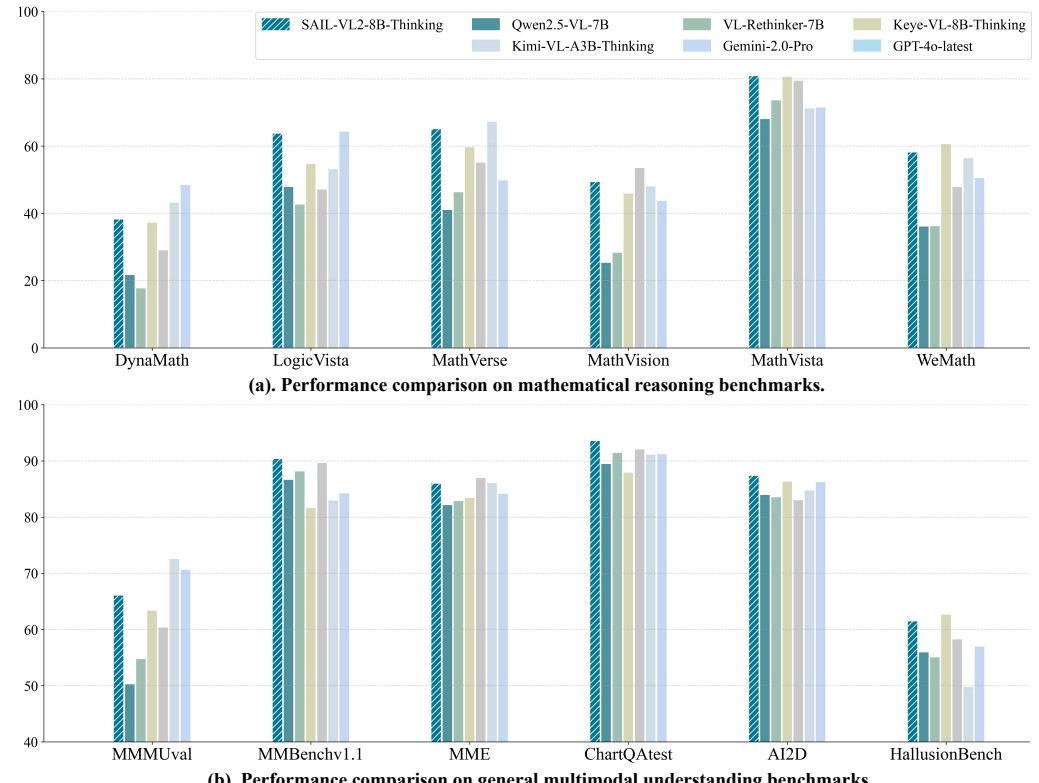

(a). Performance comparison on mathematical reasoning benchmarks.

(b). Performance comparison on general multimodal understanding benchmarks.

Figure 1: **Performance comparison between SAIL-VL2-Thinking (SAIL-VL2 post-trained with our SAIL-RL) and other LVMs.** SAIL-VL2-Thinking achieves clear advantages on both general understanding and mathematical reasoning benchmarks, surpassing open-source models at the 8B scale and delivering competitive performance against large-scale closed-source models.

Second, during optimization, models may occasionally reach correct answers through flawed or fabricated reasoning paths. Such spurious alignments are nevertheless reinforced as positive outcomes, fostering a form of "false correctness" that undermines both robustness and reliability.

Overthinking the easy, underthinking the hard: Most approaches apply the same reasoning process to all tasks, regardless of complexity. This uniformity often leads to overthinking on simple problems, introducing unnecessary cost and noisy reasoning chains. As illustrated in Figure 2, models frequently generate redundant reasoning for trivial queries (*e.g.*, object color recognition), highlighting the inefficiency of static strategies. Conversely, on complex problems, the same rigidity causes underthinking, producing shallow reasoning and inaccurate answers. The lack of adaptive control prevents models from allocating cognitive resources efficiently, unlike humans who naturally adjust their effort based on task difficulty.

To address these challenges, we propose **SAIL-RL**, a novel RL-based post-training framework for MLLMs. SAIL-RL follows the standard two-stage paradigm, consisting of a CoT–augmented SFT stage and an RL-tuning stage, but introduces a dual reward system that explicitly supervises both *the quality of reasoning* and *the adaptivity of thinking strategies*. The **Thinking Reward** moves beyond outcome-only supervision by directly assessing the reasoning process. It evaluates factual grounding to penalize hallucinations, logical coherence to ensure consistency across steps, and answer consistency to guarantee that outputs are faithfully derived from the reasoning trace. This process-level supervision explicitly couples sound reasoning with correct answers. The **Judging Reward** enhances adaptivity by enabling models to decide when deep reasoning is necessary. The model learns to adopt a direct-answer mode for simple tasks and a full reasoning mode for complex ones, improving efficiency while aligning cognitive resource allocation more closely with human behavior. Together, these two reward systems allow SAIL-RL to strengthen both the reliability and efficiency of MLLMs in reasoning and comprehensive tasks.

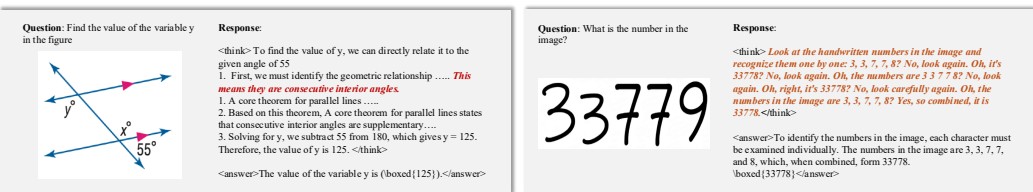

Figure 2: Limitations of current MLLMs in reasoning. **Left:** Lucky success where the model reaches the correct answer through a flawed reasoning process. **Right:** Overthinking where the model applies a needlessly complex reasoning process to a simple problem, resulting in an incorrect answer.

We conduct extensive experiments to evaluate the effectiveness of SAIL-RL. Building on the state-of-the-art MLLM SAIL-VL2, we develop SAIL-VL2-Thinking through our RL-based post-training strategy. As shown in Fig. 1, with the dual reward system, SAIL-VL2-Thinking delivers consistent gains over the baseline and conventional RL-based approaches, achieving state-of-the-art results on multiple reasoning benchmarks at 8B scales. It also reaches leading performance on OpenCompass, maintains competitive accuracy on general multimodal understanding tasks, and substantially reduces hallucinations, highlighting the robustness and reliability introduced by SAIL-RL. Together, these contributions establish SAIL-RL as a principled post-training framework that strengthens both the quality and adaptivity of reasoning in MLLMs.

## 2 SAIL-RL

In this section, we introduce SAIL-RL, a reasoning-incentivized tuning framework to improve both the effectiveness and efficiency of RL post-training for MLLMs. SAIL-RL introduces a dual-reward mechanism that guides models on *what to think* and *when to think*, thereby enhancing the quality and efficiency of reasoning. By directly addressing the above two limitations of existing RL-tuning approaches, SAIL-RL establishes a new paradigm for MLLM post-training.

### 2.1 THINKING REWARD: "WHAT TO THINK"

As the saying goes, *"sound reasoning leads to correct answers."* To improve response quality, a model is required to learn *what to think* by constructing clear and coherent reasoning paths. Beyond outcome-only supervision in conventional RL-tuning, we introduce a **Thinking Reward** that comprehensively evaluates reasoning quality with LLM-based judge models. This reward is integrated into RL tuning to guide models toward producing higher-quality reasoning across multiple dimensions.

**Logical Coherence Reward.** We first introduce the Logical Coherence Reward, which evaluates whether a model can *think clearly*. This dimension measures the internal logical integrity of the reasoning process, ensuring arguments are both well-structured and correctly executed. To this end, the judge model applies two sequential checks: (i) Structural Soundness, assessing whether the problem is properly formulated (e.g., into a valid equation); and (ii) Deductive Soundness, verifying that subsequent steps are free of contradictions, calculation errors, or logical fallacies. Failure in either check yields a score of $d_1 = 0$, and success in both yields $d_1 = 1$.

**Factual Grounding Reward.** We then introduce the Factual Grounding Reward to evaluate whether the model is *thinking truthfully* rather than hallucinating. This reward penalizes unsupported statements by requiring each step in the reasoning process to be factually grounded. To this end, the judge model performs a hierarchical fact-check across three sources: (i) Visual Grounding, verifying claims against the provided image; (ii) Textual Grounding, checking consistency with the input query; and (iii) World Knowledge, consulted only when verification is not possible from the first two sources. Any contradiction at any stage yields a score of $d_2 = 0$, and otherwise $d_2 = 1$.

**Answer Consistency Reward.** We further introduce the Answer Consistency Reward to evaluate whether the model *thinks correctly*. This reward ensures that the final answer is a direct and faithful conclusion derived from the preceding reasoning. The judge model verifies that the reasoning trace fully justifies the answer, checking for contradictions, reliance on unstated information, or unsupported claims. Any failure results in a score of 0; otherwise, the score is 1.

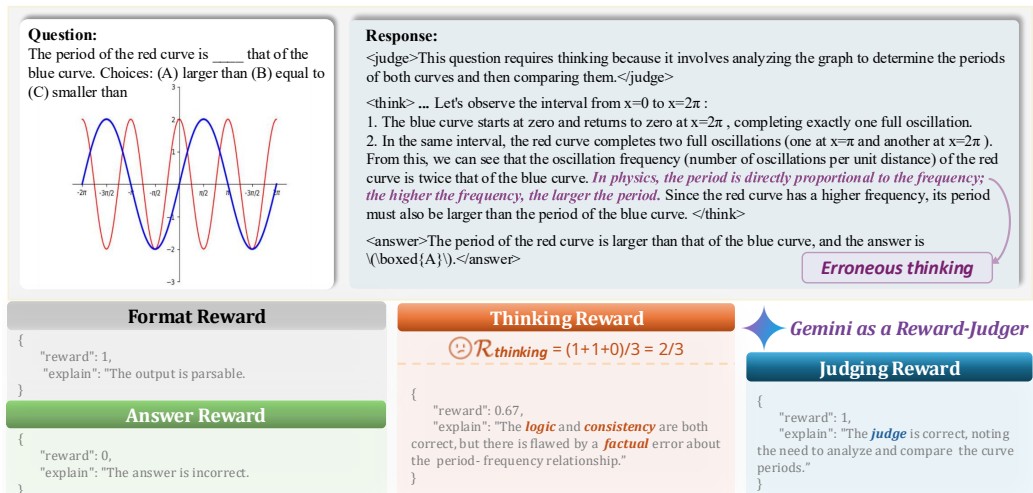

Figure 3: An overview of the SAIL-RL's multi-dimensional reward system. The system evaluates a model's response across four dimensions: Format, Answer, Thinking, and Judging. The nuanced semantic rewards for Thinking and Judging are provided by Gemini acting as a reward-judger.

Finally, the overall thinking reward $\mathcal{R}_{\text{think}}$ is computed as the average of the three binary dimensions evaluated by the judge model, *i.e.*, $\mathcal{R}_{\text{think}} = \frac{1}{3} \sum_{i=1}^{3} d_i$.

## 2.2 JUDGING REWARD: "WHEN TO THINK"

We further explore how to guide MLLMs on *when to think*: applying thorough reasoning for complex problems while giving direct responses to simple ones. The central challenge is to assess task difficulty and adapt the reasoning strategy accordingly, ensuring efficiency without sacrificing quality. To this end, we introduce a **Judging Reward**, which incentivizes the model to first determine whether reasoning is necessary before generating a response, thereby balancing effectiveness with efficiency.

Specifically, the model is required to decide whether a given question necessitates a detailed reasoning process (`think`) or can be answered directly (`direct`). The reward depends on the appropriateness of this decision, measured against ground-truth labels indicating task complexity. For complex questions, choosing `think` yields a score of $d_{\text{judge}} = 1$, and otherwise 0. For simple questions, choosing `direct` is rewarded with $d_{\text{judge}} = 1$, while initiating unnecessary reasoning results in 0. The final judging reward is therefore defined as $\mathcal{R}_{\text{judge}} = d_{\text{judge}}$. By optimizing this reward, SAIL-RL trains the model to assess the necessity of reasoning before responding, thereby improving efficiency without sacrificing accuracy on complex tasks.

## 2.3 REWARD SYSTEM

In addition to our dual-reward mechanism, we also incorporate traditional RL-tuning rewards (Shao et al., 2024). The answer reward $\mathcal{R}_{\text{answer}}$ evaluates the correctness of the final response with a binary score (1 if correct, 0 otherwise). For verifiable tasks (*e.g.*, math problems), correctness is determined programmatically through exact matching, numerical evaluation with tolerance, or execution-based validation. For open-ended tasks (*e.g.*, multimodal benchmarks), correctness is judged by a strong LLM along with factual accuracy, semantic relevance, and completeness. The format reward enforces machine-parsable outputs, requiring judgement to be enclosed within `<judge>` tags, reasoning within `<think>` tags and the final answer in a `\boxed{}` tag; any violation yields a score of 0. The overall reward signal unifies these components into a single formulation:

$$\mathcal{R}_{\text{total}} = \alpha \cdot (\mathcal{R}_{\text{judge}} \cdot \mathcal{R}_{\text{think}} \cdot \mathcal{R}_{\text{answer}}) + (1 - \alpha) \cdot \mathcal{R}_{\text{format}}, \tag{1}$$

Here, $\mathcal{R}_{\text{judge}}$, $\mathcal{R}_{\text{think}}$, $\mathcal{R}_{\text{answer}}$, and $\mathcal{R}_{\text{format}}$ denote the respective rewards. The combination of $\mathcal{R}_{\text{judge}}$, $\mathcal{R}_{\text{think}}$, and $\mathcal{R}_{\text{answer}}$ is designed as a cascading product. An additive approach is susceptible to reward hacking, where a model could be rewarded for success in one component despite critical failures in

another. For example, this could reward a *lucky guess* (a correct answer from flawed reasoning) or *overthinking* (generating a high-quality but unnecessary reasoning for a simple question). In contrast, our cascading product formulation functions as a logical AND gate. It ensures that a reward is given only when the initial judgment, the reasoning process, and the final answer are all correct. This design effectively mitigates these undesirable behaviors, thereby enforcing a strong link between sound reasoning and correct outcomes. The $\mathcal{R}_{\text{format}}$ serves as a lighter regularizer to encourage structural compliance. We set $\alpha = 0.9$ to emphasize correctness and reasoning quality.

## 2.4 POST-TRAINING STRATEGY

### 2.4.1 LONGCOT SFT

The first stage builds the model's foundational ability to sequentially judge a problem's complexity, generate a step-by-step reasoning process, and derive a final answer. This is achieved by fine-tuning the base model on a large-scale, high-quality LongCoT dataset.

**Data Curation.** The dataset is built through a comprehensive pipeline. We aggregate data from diverse sources (Jia et al., 2025; Shi et al., 2024; Xu et al., 2024), followed by rigorous cleaning and deduplication. We unify the data into a structured format that explicitly includes judgment, thinking, and answering tags. To generate this data, a powerful teacher model is prompted to first produce a judgment (`<judge>`) on whether the problem requires complex reasoning. It then generates a detailed thinking process (`<think>`) that logically leads to the ground-truth answer, which is enclosed in the `\boxed{}` tag. The generated samples are then filtered through multi-stage checks for redundancy, correctness, and reasoning complexity. This process yields 400K high-quality LongCoT samples. Further implementation details are provided in the Section. 3.

**Training Objective.** We fine-tune the model with AdamW (batch size 1024) and a cosine learning rate peaking at 1e-6. The objective is a standard next-token prediction loss over the full sequence of judgment, reasoning, and answer:

$$\mathcal{L}_{\text{LongCoT-SFT}} = -\frac{1}{|D_{\text{CoT}}|} \sum_{(I,J,T,A) \in D_{\text{CoT}}} \log P_\theta(J \circ T \circ A | I) \tag{2}$$

where $I$ is the input, $J$ is the judgment text, $T$ the reasoning process, $A$ the final answer, and $\circ$ denotes concatenation. This training objective explicitly teaches the model to first judge the problem's nature, then produce a corresponding reasoning trace, and finally output the answer in the correct format.

### 2.4.2 RL TUNING WITH REWARD SYSTEM

LongCoT SFT provides a strong generative template, RL incentivizes reasoning capabilities. It focuses on not only what to think (reasoning quality) but also when to think (efficiency), thereby ensuring sound reasoning consistently leads to correct answers.

**Data Curation.** We curate data from diverse sources, including Math (Sun et al., 2024; Meng et al., 2025), Puzzle (Chia et al., 2024), Science (Wang et al., 2025; Lu et al., 2022), OCR (Chen et al., 2025b), and Counting (Johnson et al., 2017). A two-stage filtering pipeline is applied: (i) multiple-choice questions are converted into free-response format to prevent reward hacking, and (ii) difficulty-based filtering with the previous-stage model's pass@4 score removes trivial and unsolvable samples. This yields a high-quality dataset of 70K problems for stable RL training.

**Training Objective.** We optimize the SFT model using DAPO (Yu et al., 2025) on the curated dataset with our proposed reward system. We remove the KL loss and set the learning rate to 1e-6. The clipping value $\varepsilon$ is dynamically adjusted within the range of [0.20, 0.28] to encourage exploration Further implementation details and hyperparameters are provided in the details in the Section. 3.

## 3 EXPERIMENTS

### 3.1 EXPERIMENT SETUP

This section details the methodology for our experiments. including training datasets, implementation details, and evaluation benchmarks.

### 3.1.1 TRAINING DATASETS

We prepare two distinct datasets for the two-stage training pipeline. First, for LongCoT SFT, we generate 400K high-quality CoT data from diverse sources. All samples are unified into the structured sequence format `<judge>...</judge><think>...</think>` and `\boxed{...}` to explicitly teach the model to "judge first, then think, and finally answer". Second, for the RL, we curate a mixed RL dataset, which consists of two parts: 50K from STEM for precise reward signals, and 20K from general tasks to enhance generalization.

**LongCoT Data Curation.**

We construct a high-quality LongCoT dataset of 400K samples in a `judge-think-answer` format, aimed to instill a meta-cognitive capabilities: first judging a problem's complexity, then executing the reasoning process and finally give the answer. We first collect diverse data, ranging from complex reasoning (e.g., VisualWebInstruct (Jia et al., 2025), MathV360K (Shi et al., 2024)) to simple perception (e.g., LLaVA-CoT (Xu et al., 2024)). All collected data are then processed through a unified pipeline as follows: **1) Data Cleaning:** We remove extraneous noise (e.g., system prompts) and deduplicate based on unique image-question pairs. **2) Conditional Annotation:** Samples are annotated by complexity. Complex problems receive detailed CoT in `<think>` with a positive `<judge>` label, while simple tasks use an empty `<think>` block (`\n\n`) with a negative label. All answers are standardized in `<boxed>`. **3) Quality Filtering:** We filter trivial reasoning via token overlap checks and balance chain lengths to ensure diversity.

**RL Data Curation.** We construct a comprehensive dataset of 70K samples for the RL stage, strategically composed of specialized STEM problems and general-purpose problems. The STEM domain, which contains 50K samples, is curated from a wide range of public benchmarks in fields such as Math (Sun et al., 2024; Meng et al., 2025), Puzzles (Chia et al., 2024), Science (Wang et al., 2025; Lu et al., 2022), OCR (Chen et al., 2025b), and Counting (Johnson et al., 2017). This data undergoes a rigorous two-stage filtering pipeline to ensure high-quality training signals. First, to mitigate reward hacking, we reformat multiple-choice questions into an open-ended, free-response format. Second, we implement a difficulty-aware filter using our SFT model's `pass@4` score, retaining only problems within an optimal learning range by discarding trivial (`pass@4=1`) and unsolvable (`pass@4=0`) instances. To maintain broad capabilities, we incorporate 20K General QA samples from LLaVA-OneVision (Li et al., 2024), filtered primarily for quality and instructional diversity. This resulting mixture provides a robust training environment, prioritizing complex reasoning while preserving general interaction abilities.

### 3.1.2 IMPLEMENTATION DETAILS

Our model, based on SAIL-VL2 (Yin et al., 2025) (integrating AimV2 (Fini et al., 2025) and Qwen3 (Yang et al., 2025a)), is trained in two stages. We first conduct full-parameter SFT for one epoch on our LongCoT dataset. This is followed by three epochs of RL using the DAPO algorithm, guided by our SAIL-RL reward system which leverages Gemini-2.5-Pro as the VLM-Judge.

**LongCoT SFT Stage.** In the first stage, we fine-tune all parameters of the model for **one epoch** on our 400K-sample LongCoT dataset. For this SFT stage, we set the maximum sequence length to 20K, the global batch size to 1024, and the learning rate to 1e-6.

**RL Stage.** Subsequently, we optimize the SFT model for three epochs on our 70K-sample mixed RL dataset using the DAPO (Yu et al., 2025) algorithm, guided by our proposed SAIL-RL reward system. We set the maximum sequence length to 20K, consisting of 16K for the input and 4K for the output. The policy learning rate is set to 1e-6 with a global PPO batch size of 256. For each sample, we rollout 5 times to estimate the advantage. To encourage exploration and stabilize training, we remove the standard KL divergence and dynamically adjust the clipping value $\varepsilon$ within the range of [0.20, 0.28] to encourage exploration.

### 3.1.3 EVALUATION BENCHMARKS

We conduct a comprehensive evaluation using VLMEvalKit (Duan et al., 2024), with GPT-4o-Mini as the judge for fairness. We assess two primary categories of abilities. First, we evaluate advanced reasoning on benchmarks focused on mathematical and logical analysis (Zou et al., 2024; Zhang et al., 2024; Lu et al., 2023; Fang et al., 2024). Second, we evaluate general multimodal understanding using

Table 1: Evaluation results on OpenCompass multimodal reasoning benchmarks. The best results among open-source models are **bolded** and the second-best results are underlined.

| Model | DynaMath | LogicVista | MathVerse | MathVision | MathVista | WeMath | Average |
|---|---|---|---|---|---|---|---|
| *Close-source Models* | | | | | | | |
| Gemini-2.0-Pro | 43.3 | 53.2 | 67.3 | 48.1 | 71.3 | 56.5 | 56.6 |
| GPT-4o-latest | 48.5 | 64.4 | 49.9 | 43.8 | 71.6 | 50.6 | 54.8 |
| *Open-source Models* | | | | | | | |
| InternVL3-2B | 14.0 | 33.6 | 20.6 | 20.2 | 57.3 | 13.0 | 26.5 |
| Qwen2.5-VL-3B | 11.0 | 36.0 | 29.3 | 18.1 | 60.2 | 20.7 | 29.2 |
| WeThink-7B | 24.4 | 53.0 | 44.7 | 27.2 | 70.9 | 48.0 | 44.7 |
| InternVL3-8B | 25.7 | 44.5 | 38.5 | 30.0 | 70.5 | 39.5 | 41.5 |
| Qwen2.5-VL-7B | 21.8 | 47.9 | 41.1 | 25.4 | 68.1 | 36.2 | 40.1 |
| VL-Rethinker-7B | 17.8 | 42.7 | 46.4 | 28.4 | 73.7 | 36.3 | 40.9 |
| VLAA-Thinker-7B | 22.4 | 48.5 | 48.2 | 26.4 | 68.0 | 41.5 | 42.5 |
| Keye-VL-8B-Thinking | 37.3 | 54.8 | 59.8 | 46.0 | 80.7 | **60.7** | 56.6 |
| Kimi-VL-A3B-Thinking | 29.1 | 47.2 | 55.2 | **53.6** | 79.5 | 47.9 | 52.1 |
| SAIL-VL2-2B-Instruct | 10.2 | 36.2 | 22.6 | 23.4 | 71.1 | 22.7 | 31.0 |
| SAIL-VL2-2B-LongCoT | | | | | | | |
| SAIL-VL2-2B-Thinking | 25.7 | 45.4 | 50.5 | 30.5 | 73.6 | 42.1 | 44.6 |
| SAIL-VL2-8B-Instruct | 17.8 | 45.0 | 32.9 | 27.6 | 76.4 | 35.8 | 39.3 |
| SAIL-VL2-8B-LongCoT | 29.7 | 58.2 | 53.1 | 39.7 | 77.2 | 54.4 | 52.1 |
| SAIL-VL2-8B-Thinking | **38.3** | **63.8** | **65.1** | 49.4 | **80.9** | 58.2 | **59.3** |

Table 2: Evaluation on multimodal understanding benchmarks. HallBench denotes HallusionBench. The best results among open-source models are **bolded** and the second-best results are underlined.

| Model | General VQA | | | OCR & Chart | | Hallucination | Average |
|---|---|---|---|---|---|---|---|
| | MMMU$_{val}$ | MMBench$_{v1.1}$ | MME | ChartQA$_{test}$ | AI2D | HallBench | |
| *Close-source Models* | | | | | | | |
| Gemini-2.0-Pro | 72.6 | 83.0 | 86.1 | 91.2 | 84.8 | 49.8 | 77.9 |
| GPT-4o-latest | 70.7 | 84.3 | 84.2 | 91.5 | 86.3 | 57.0 | 79.0 |
| *Open-source Models* | | | | | | | |
| InternVL3-2B | 47.1 | 84.3 | 77.4 | 80.4 | 78.7 | 41.4 | 68.2 |
| Qwen2.5VL-3B | 48.1 | 82.4 | 77.5 | 87.0 | 80.7 | 48.3 | 70.7 |
| WeThink-7B | 50.9 | 87.8 | 82.9 | 90.8 | 84.5 | 55.1 | 75.3 |
| InternVL3-8B | 57.3 | 87.7 | 85.2 | 89.6 | 85.2 | 53.7 | 76.5 |
| Qwen2.5-VL-7B | 50.3 | 86.7 | 82.2 | 89.5 | 84.0 | 56.0 | 74.8 |
| VL-Rethinker-7B | 54.8 | 88.2 | 82.9 | 91.5 | 83.6 | 55.1 | 76.0 |
| VLAA-Thinker-7B | 51.9 | 86.9 | 83.3 | 89.5 | 78.9 | 51.5 | 73.7 |
| Keye-VL-8B-Thinking* | 63.4 | 81.7 | 83.5 | 88.0 | 86.4 | **62.7** | 77.6 |
| Kimi-VL-A3B-Thinking* | 60.4 | 89.7 | **87.0** | 92.1 | 83.1 | 58.3 | 78.4 |
| SAIL-VL2-2B-Instruct | 47.7 | 86.8 | 76.6 | 89.1 | 83.0 | 51.7 | 72.5 |
| SAIL-VL2-2B-LongCoT | 44.6 | 82.5 | 74.7 | 90.2 | 77.4 | 54.0 | 70.6 |
| SAIL-VL2-2B-Thinking | 51.2 | 87.2 | 78.4 | 92.2 | 84.1 | 53.1 | 74.1 |
| SAIL-VL2-8B-Instruct | 55.4 | 90.2 | 84.5 | 90.3 | **87.7** | 55.1 | 77.2 |
| SAIL-VL2-8B-LongCoT | 63.0 | 88.7 | 82.6 | 91.3 | 83.6 | 59.4 | 78.1 |
| SAIL-VL2-8B-Thinking | **66.1** | **90.4** | 86.0 | **93.6** | 87.4 | 61.5 | **80.8** |

a comprehensive benchmarks covering from general VQA, chart comprehension to hallucination detection (Yue et al., 2024; Liu et al., 2024).

## 3.2 MAIN RESULTS

### 3.2.1 BENCHMARK PERFORMANCE

**Multimodal Reasoning Benchmarks.** As shown in Tab. 1, SAIL-VL2-8B-Thinking sets a new state-of-the-art among open-source models with an average score of 59.3. This gain is largely attributed to the thinking reward, which supervises the reasoning process rather than only the final answer, leading to a +20.0 improvement over the baseline SAIL-VL2-8B (39.3). The model achieves top-tier results

on complex tasks such as DynaMath (38.3), LogicVista (63.8), MathVerse (65.1), and MathVista (80.9), surpassing even closed-source systems like GPT-4o (54.8) and Gemini-2.0-Pro (56.6).

**Multimodal Understanding Benchmarks.** As shown in Tab. 2, SAIL-VL2-8B-Thinking achieves an open-source state-of-the-art with an average score of 80.4. The thinking quality reward improves factual grounding, reducing hallucinations as reflected in the 61.5 score on HallusionBench, while the judge reward allocates cognitive effort effectively, achieving 93.6 on ChartQA. The synergy of these mechanisms enhances both reasoning quality and efficiency, ensuring robust performance.

### 3.2.2 THINKING ANALYSIS

**Thinking Quality.** To validate the effectiveness of the proposed thinking reward, we analyze the reasoning chains generated by SAIL-VL2-8B-Thinking and compare them with leading open-source models that also produce explicit thinking steps, such as Keye-VL-8B-Thinking. As shown in Tab. 3, we achieves higher thinking quality across multiple dimensions, with average scores of 83.2 on LogicVista and 95.5 on OCRBench, surpassing Keye-VL-8B-Thinking. These results indicate that the thinking quality reward not only encourages models to generate reasoning steps but also guides them to reason correctly, leading to more reliable and accurate outcomes.

Table 3: Evaluation results on thinking quality. We use the proposed reward system to compare SAIL-VL2-8B-Thinking and Keye-VL-8B-Thinking.

| Benchmark | Model | Logic | Hallucination | Consistency | Average |
|---|---|---|---|---|---|
| | | Score (%) | Score (%) | Score (%) | |
| LogicVista | SAIL-VL2-8B-Thinking | 80.3 | 73.8 | 95.3 | 83.2 |
| | Keye-VL-8B-Thinking | 55.3 | 61.7 | 78.8 | 65.3 |
| | $\Delta$ (Improvement) | +25.1 | +12.1 | +16.5 | +17.9 |
| OCRBench | SAIL-VL2-8B-Thinking | 95.1 | 94.0 | 97.4 | 95.5 |
| | Keye-VL-8B-Thinking | 89.9 | 87.6 | 87.4 | 88.3 |
| | $\Delta$ (Improvement) | +5.2 | +6.4 | +10.0 | +7.2 |

**Thinking Trigger.** To validate the effectiveness of our proposed judging reward, we analyze the adaptive allocation of reasoning resources in the model. As shown in Fig. 4, our model intelligently adjusts its thinking trigger rate based on the task type, demonstrating notable efficiency and rationality. For example, on tasks like OCR-Bench where complex reasoning is often unnecessary, the trigger rate is merely 7.5%. In contrast, for tasks requiring deep mathematical or logical inference like MathVista and We-Math, the rate significantly increases to 94% and 99.1%, respectively. Crucially, this efficiency does not come at the cost of performance. As demonstrated in Tab. 1 and Tab. 2, SAIL-VL2-8B-Thinking achieves superior results while be-

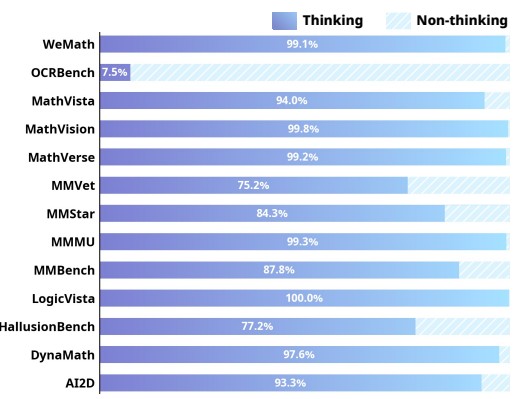

Figure 4: Evaluation results on thinking trigger.

ing significantly more efficient than models with non-adaptive thinking mechanisms. This provides strong evidence that our Thinking Judge can effectively discern task complexity, activating deep reasoning only when necessary to achieve an optimal balance between performance and efficiency.

### 3.3 ABLATION STUDY

Due to the computational cost of the full training, we conduct our experiments on a smaller model, SAIL-VL2-2B. We also adopt a shortened training schedule, training for a single epoch at each stage.

### 3.3.1 Effectiveness of Thinking Reward

To validate the effectiveness of Thinking Reward, we compare two variants: (1) a baseline trained with an answer reward that only considers final correctness, and (2) our model trained with the proposed reward, which additionally supervises intermediate reasoning steps.

**Performance Comparison.** As shown in Tab. 4, Thinking Reward yields a +1.3% average gain across eight benchmarks. We show significant improvement in STEM tasks that require multi-step reasoning, for example, achieving a **+2.5%** gain on WeMath. Although MME has a -0.4% minor drop, the overall positive results demonstrate that complementing answer rewards with thinking reward produces more reliable reasoning and accurate answers.

Table 4: Ablation on Thinking Reward. We compares answer reward against thinking reward.

| Reward | STEM | | | | General | | | Hallucination | Average |
|---|---|---|---|---|---|---|---|---|---|
| | WeMath | MathVerse | LogicVista | DynaMath | MMMU$_{val}$ | MMBench$_{v1.1}$ | MME | HallBench | |
| Answer | 38.7 | 47.8 | 45.0 | 22.6 | 46.9 | 84.5 | 77.5 | 51.7 | 51.8 |
| Answer + Thinking | 41.2 | 49.5 | 47.1 | 25.4 | 47.4 | 85.3 | 77.1 | 52.1 | **53.1** |
| Δ (Improvement) | +2.5 | +1.7 | +2.1 | +2.8 | +0.5 | +0.8 | -0.4 | +0.4 | **+1.3** |

**Training Dynamics.** Fig. 5 shows the evolution of three quality metrics—logic, consistency, and hallucination—during training. Our model steadily improves across all dimensions, while the baseline stagnates, and even degrades on answer consistency. This provides direct evidence that optimizing for final answers alone does not guarantee, and may even harm, coherent reasoning.

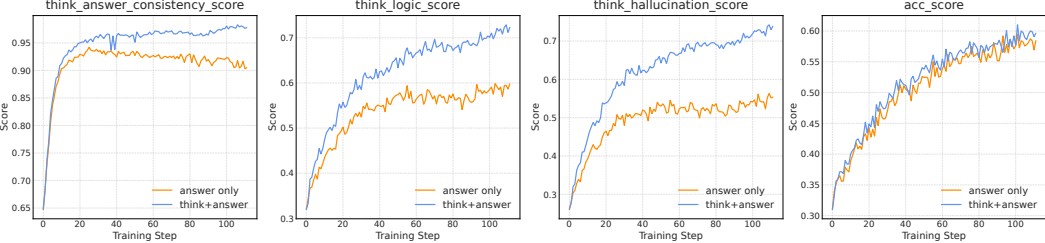

Figure 5: **Ablation on training dynamics of thinking reward.** Our method (blue) consistently improves all three thinking score over the answer-only baseline (orange), which stagnates or degrades.

**Case Study.** As shown in Fig. 7, the thinking reward significantly improve the cognitive depth. The baseline model (blue), guided by an answer-only reward, adopts a brute-force strategy. It correctly computes the series term-by-term until the condition is met at n=4 and then immediately terminates its reasoning. While effective for finding the answer, this approach demonstrates a superficial understanding of the problem. In contrast, our model (orange), trained with the thinking reward, exhibits a more advanced analytical process. After finding the solution, it continues to analyze the series, successfully identifying the underlying pattern of alternating signs and increasing magnitudes. This demonstrates that by rewarding the quality of the reasoning process itself, our approach encourages the model to move beyond simple procedural execution towards a deeper, more human-like understanding of the problem's structure.

### 3.3.2 Effectiveness of Judging Reward

We conduct an ablation study comparing two reward strategies: a Forced Thinking baseline, which consistently engages in step-by-step reasoning, and a Judge Reward approach, which dynamically decides whether to activate the thinking process.

**Performance Comparison.** As shown in Tab. 5, our Judge Reward yields a +0.6% average gain. The model maintains competitive performance on STEM benchmarks, confirming its core reasoning abilities are unaffected. The advantage is concentrated in general and perception-heavy tasks where avoiding overthinking is crucial. For example, we achieves significant gains on MMBench (+1.3%), and MME (+0.5%), reducing factual errors and analytical confusion. These results validate that Judging Reward dynamically allocates resources to achieve robust reasoning.

Table 5: Ablation on Judging Reward. We compare forced thinking against judging reward.

| Method | STEM | | | | General | | | Hallucination | Average |
|---|---|---|---|---|---|---|---|---|---|
| | WeMath | MathVerse | LogicVista | DynaMath | MMMU$_{val}$ | MMBench$_{v1.1}$ | MME | HallBench | |
| Forced Thinking | 38.7 | 47.8 | 45.0 | 22.6 | 46.9 | 84.5 | 77.5 | 51.7 | 51.8 |
| Judge Reward | 38.5 | 47.9 | 45.3 | 22.7 | 48.3 | 85.8 | 78.0 | 53.1 | 52.4 |
| $\Delta$ (Improvement) | -0.2 | +0.1 | +0.3 | +0.1 | +1.4 | +1.3 | +0.5 | +1.4 | +0.6 |

**Case Study.** Fig. 6 highlights how our Judging Reward prevents errors from overthinking. The baseline model (blue), forced to generate a reasoning chain, over-analyzes a simple OCR task, leading to hallucinations ("tongue") and an incorrect answer. In contrast, our model (orange box) uses the Judge Reward to identify the task as straightforward, bypass the thinking process, and answer correctly. This demonstrates the reward's critical role in dynamically allocating cognitive resources to avoid errors on simple perception tasks.

## 4 RELATED WORK

**Cognitive Paradigms in MLLMs.** Recent advancements in MLLMs can be categorized into two distinct cognitive paradigms: System 1 and System 2. System 1 models, exemplified by state-of-the-art generalist models like GPT-4o (Hurst et al., 2024) and Qwen-2.5-VL (Bai et al., 2025), prioritize rapid, intuitive visual perception. These models typically utilize direct instruction tuning to map visual features to textual outputs, excelling at perception-intensive tasks but often struggling with complex logical derivations. Conversely, System 2 models aim to emulate slow, deliberative reasoning. Following the breakthrough of DeepSeek-R1 (Guo et al., 2025) in text reasoning, recent works have extended this paradigm to the multimodal domain. Models such as Gemini-2.5-Pro (Comanici et al., 2025) and Kimi-VL-Thinking (Team et al., 2025a) leverage reinforcement learning to internalize extensive Chain-of-Thought (CoT) processes. By generating explicit thinking tokens before the final answer, these models achieve significant gains in complex visual math and logic tasks. However, the high inference cost of System 2 reasoning necessitates dynamic architectures. Recent works (Zhang et al., 2025; Zhou et al., 2025) have begun exploring routing mechanisms to switch between these modes, yet optimizing when and how to switch remains an open challenge.

**Multimodal RL.** RL has emerged as a powerful paradigm for enhancing long-chain reasoning in LLMs (OpenAI, 2024; Guo et al., 2025). Leveraging this foundation, recent research has extended RL to MLLMs (Chen et al., 2025a; Team et al., 2025b; Deng et al., 2025; Wang et al., 2025), achieving significant improvements in visual reasoning. However, directly applying these RL methods to multimodal tasks introduces two critical challenges: efficiency bottlenecks caused by over-thinking simple multimodal questions, and effectiveness bottlenecks caused by incorrect multimodal reasoning chains. Consequently, current optimization strategies largely focus on addressing these symptoms in isolation. To mitigate efficiency issues, works such as R-4B employ a bi-mode policy optimization to determine whether to activate the reasoning process, while FAST-GRPO (Xiao et al., 2025) introduces adaptive length penalty to curb redundant steps. To improve reasoning quality, VisualPRM (Wang et al.) and URSA (Luo et al.) utilize visual process reward models for step-by-step reasoning verification. In contrast to these approaches that typically optimize dimensions in isolation, we aim to bridge this gap by designing a dual-reward mechanism, which simultaneously learns when to reason and how to reason, unifying efficiency and effectiveness within a single framework.

## 5 CONCLUSION

In this paper, we propose SAIL-RL, a reinforcement learning post-training framework that advances the reasoning capability of MLLMs by teaching them when and how to think. Unlike outcome-only supervision and uniform reasoning strategies, SAIL-RL introduces a dual reward system that evaluates reasoning quality and adaptively controls the depth of thinking. Experiments on SAIL-VL2 show that SAIL-RL improves reasoning and multimodal understanding benchmarks at both 4B and 8B scales, achieves state-of-the-art results among models of comparable size, and delivers competitive performance with GPT-4o and Gemini-2.0-Pro. These findings establish SAIL-RL as a foundation that supports the design of more reliable and adaptive MLLMs through scalable thinking RL.

# 6 ETHICS AND REPRODUCIBILITY STATEMENT

## 6.1 ETHICS STATEMENT

We strictly follow the ICLR Code of Ethics.

**Data, Bias, and Safety.** This research is based entirely on publicly available academic benchmark datasets, and we strictly adhere to their original licenses. We did not collect any personally identifiable information (PII) or biometric data during our research. While our work aims to reduce hallucinations by enhancing logical coherence and factual grounding, we acknowledge that as a framework built upon large pre-trained models, the model may still generate harmful, biased, or unsafe content. Our system does not directly use sensitive personal attributes, but biases originating from the pre-training data may still exist. We strive to identify and quantify these biases by evaluating on a diverse set of benchmarks.

**Fairness, Environment, and Disclosure.** We will disclose all funding sources and author affiliations in the camera-ready version of the paper. Sponsors had no influence on our research. Our research primarily consumes computational resources, and we endeavor to reduce energy consumption by optimizing training pipelines and employing efficient model architectures. To promote advancement in the field, we commit to publicly releasing our code, model checkpoints, reward function implementations, and detailed documentation with appropriate licenses upon acceptance, in line with the ICLR Code of Ethics.

## 6.2 REPRODUCIBILITY STATEMENT

We are committed to maximizing the reproducibility of our research.

**Architectures and Hyperparameters:** The main paper details our model architecture, the SAIL-RL training pipeline, the implementation details of the dual reward system (*Thinking Reward* and *Judging Reward*), and all key hyperparameters.

**Datasets and Evaluation:** We clearly list all public benchmark datasets used (e.g., OpenCompass) and provide a detailed description of the evaluation process. The appendix contains further details on task setups, data preprocessing, and qualitative analysis (e.g., case studies of the model's thinking process).

**Code and Models:** Upon acceptance, we will publicly release the complete codebase and the model checkpoints trained on SAIL-VL2 to facilitate reproduction and future research by the community.

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

## A  PRELIMINARIES

In this section, we review the reasoning-incentivized MLLM post-training pipeline of existing RL-tuning methods, which serves as the preliminaries for SAIL-RL. The reasoning-incentivized MLLM post-training pipeline typically consists of two stages: supervised fine-tuning (SFT) on complex reasoning datasets to elicit long chain-of-thought (CoT) capability, followed by reinforcement learning (RL) with verifiable rewards to further strengthen reasoning.

**Stage 1: LongCoT SFT.** The first stage trains the model to learn step-by-step reasoning structures through SFT. Each sample includes a reasoning trace ($T$) leading to a final answer ($A$), and the objective is standard next-token prediction over the concatenated sequence:

$$\mathcal{L}_{\text{SFT}} = -\mathbb{E}_{(I,T,A)\sim D_{\text{CoT}}}\big[\log P_\theta(T \circ A|I)\big].$$

This equips the model with the ability to articulate reasoning paths.

**Stage 2: RL with Verifiable Rewards (RLVR).** The second stage enhances reasoning with reinforcement learning. The reward $\mathcal{R}$ combines final-answer accuracy and output-format compliance:

$$\mathcal{R} = \alpha \cdot \mathcal{R}_{\text{answer}} + (1 - \alpha) \cdot \mathcal{R}_{\text{format}}.$$

Policy optimization is performed with Group Relative Policy Optimization (GRPO), which computes normalized advantages across peer responses and updates the policy via a clipped surrogate objective regularized by a KL term:

$$\mathcal{J}_{\text{GRPO}}(\theta) = \mathbb{E}\Big[\min\big(r_t(\theta)\hat{A}_t, \text{clip}(r_t(\theta), 1 \pm \varepsilon)\hat{A}_t\big) - \beta D_{\text{KL}}(\pi_\theta || \pi_{\text{ref}})\Big],$$

where $r_t(\theta) = \frac{\pi_\theta}{\pi_{\theta_{\text{old}}}}$. This stage directly aligns responses with both structural and answer correctness.

## B  ADDITIONAL EXPERIMENTS

### B.1  ABLATION ON REWARD FORMULATION

We adopt the cascading product formulation to strictly align the judge, reasoning, and answer components. For comparison, we implement an *additive baseline* where the rewards are calculated

as the arithmetic mean ($R_{\text{total}} = \frac{1}{3} \sum R_i$), which treats each component independently and allows for partial credit. Tab. 6 presents the ablation results. We observe that the **Cascading Product** strategy significantly outperforms the Additive Combination across all benchmarks (e.g., **+3.5%** on MathVision). This performance gap confirms that the additive approach suffers from "score compensation"—where the model masks reasoning errors by greedily optimizing the final answer— whereas our cascading product acts as a **logical gate**, enforcing the generation of a fully coherent reasoning chain.

Table 6: Ablation study on Reward Aggregation. We compare our cascading product mechanism against the additive combination.

| Reward Aggregation | Benchmarks | | | Average |
|---|---|---|---|---|
| | **MathVision** | **LogicVista** | **MMMU$_{\text{val}}$** | |
| Additive Combination | 46.1 | 60.7 | 63.8 | 56.9 |
| Cascading Product (Ours) | **49.6** | **63.8** | **66.1** | **59.8** |
| $\Delta$ (Improvement) | +3.5 | +3.1 | +2.3 | +2.9 |

We further investigate the impact of reward signal granularity by comparing continuous scalar scores ($0 \sim 1$) against discrete binary outcomes ($0/1$). As shown in Tab. 7, the discrete setting consistently outperforms the continuous approach (e.g., **+3.2%** on Average), primarily because VLM-based judges often struggle to produce consistent fine-grained likelihoods. Consequently, binary rewards provide a sharper, unambiguous learning signal that mitigates calibration noise and guides the policy more effectively towards correct reasoning paths.

Table 7: Ablation study on Reward Signal Type. We compare using discrete binary rewards ($0/1$) against continuous scalar rewards ($0 \sim 1$). Discrete signals provide sharper guidance for RL training.

| Reward Signal | Benchmarks | | | Average |
|---|---|---|---|---|
| | **MathVision** | **LogicVista** | **MMMU** | |
| Continuous ($0 \sim 1$) | 47.5 | 59.4 | 62.8 | 56.6 |
| Discrete ($0/1$) | **49.6** | **63.8** | **66.1** | **59.8** |
| $\Delta$ (Improvement) | +2.1 | +4.4 | +3.3 | +3.2 |

## B.2 ABLATION ON REWARD MODEL SELECTION

We investigate the generalization and robustness of our framework by evaluating its performance when trained with different VLMs as the reward model. We experiment with the proprietary GPT-5 and the open-weights Qwen2.5-VL-32B. As shown in Tab. 8, the results yield two key insights. First, the proposed RL framework consistently and substantially outperforms the SFT baseline, regardless of the reward model used. For instance, the average score jumps from 42.7% to 59.9% with RL, demonstrating the robust effectiveness of our dual-reward mechanism. Moreover, our framework is not overly sensitive to the strength of the reward model. The performance gap between using the top-tier GPT-5 and the open-weights Qwen2.5-VL-32B is remarkably small, indicating that our method does not strictly depend on access to the most powerful proprietary models but can effectively learn from a variety of "teachers" to achieve strong results.

Table 8: Ablation study on Reward Model Selection. We compare the performance using different VLMs as the Judge. All RL variants significantly outperform the SFT baseline.

| Training Setup | Reward Model | Benchmarks | | | Average |
|---|---|---|---|---|---|
| | | **MathVision** | **LogicVista** | **MMMU$_{\text{val}}$** | |
| **SFT (Baseline)** | – | 27.6 | 45.0 | 55.4 | 42.7 |
| **RL with VLM-as-Judge** | Qwen2.5-VL-32B | 48.4 | 62.7 | 64.9 | 58.7 |
| | GPT-5 | **49.7** | 63.5 | **66.4** | **59.9** |
| | Gemini-2.5-Pro (Ours) | 49.6 | **63.8** | 66.1 | 59.8 |

## C    CASE STUDY

We present two case studies that highlight the distinct benefits of our proposed reward components. The first case, a math problem, demonstrates how our **Thinking Reward** encourages the model to move beyond superficial, procedural answer-finding towards a deeper, analytical understanding of a problem's structure. The second case, a simple OCR task, illustrates how the **Judging Reward** provides the model with a critical meta-cognitive ability to avoid "overthinking" by dynamically assessing task complexity and allocating its reasoning resources effectively. Detailed visualizations and analyses for both cases are provided in the Fig. 6 and Fig. 7, respectively.

| **Question**: what is written in the image? | **Forced Thinking** | **Judge Reward** |
|---|---|---|
| | \<think\> Got it, let's see. The task is to figure out what's written in the image. The input image shows some text. Let me look at it again. The image has large, cursive blue text. The main word at the top is "Fear". Then below it, there's another word, "tongue". So putting them together, the text in the image is "Fear tongue". The original answer provided is "Fear tongue", which matches what's visible. So the answer should be that.\</think\> | \<judge\>This question does not require thinking. It is a straightforward perception task that requires Optical Character Recognition (OCR) to identify and transcribe the text from the image, rather than complex reasoning.\</judge\> |
| | | \<think\>\\n\\n\</think\> |
| | The image contains the text "Fear tongue".\\boxed{Fear tongue} | The image contains the text "Fear tongue".\\boxed{Fear} |

Figure 6: Visualizing behavior on an OCR task under two different reasoning strategies. **Orange**: The output from a baseline that is forced to think. **Blue**: The output from our model guided by the proposed Judge Reward, which dynamically decides when to think.

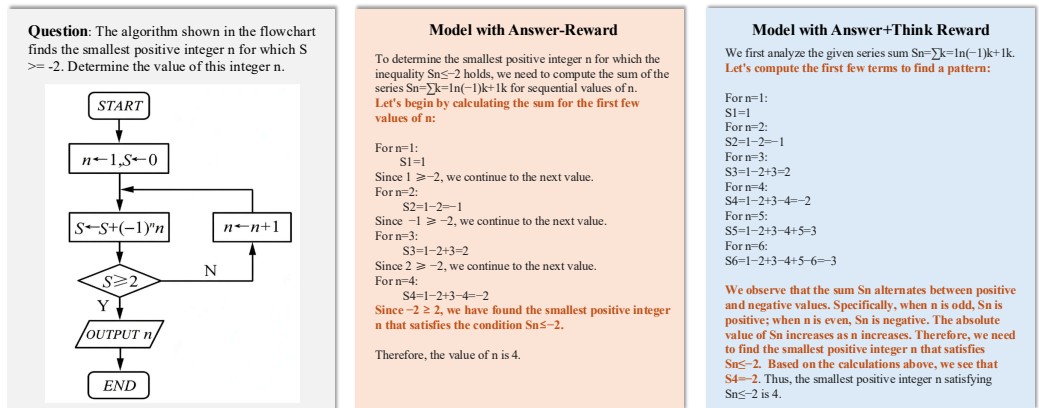

Figure 7: Visualizing reasoning on a math problem under two different reward systems. **Orange**: The output from a baseline trained with an answer-only reward. **Blue**: The output from our model trained with the proposed Thinking Quality Reward.

## D    REWARD PROMPT

This section details the set of thinking and judge reward prompts during the training of SAIL-RL. These thinking reward prompts are designed to assess thinking quality such as logical consistency, factual accuracy (hallucination), and the structural soundness of reasoning. Additionally, it includes a judge reward prompt to evaluate a model's meta-cognition, specifically its ability to determine if a question requires reasoning.

**Judge Reward Prompt**

**1. ROLE AND GOAL**
You are a top-tier AI logic analyst. Your task is to generate a ground truth assessment of task complexity and evaluate whether a "model to be evaluated" has correctly judged if a question requires "reasoning".

To do this, you will perform two core tasks:

1. **Independent Assessment**: Analyze the [QUESTION] and [IMAGE] yourself to determine if the question truly requires reasoning (Complex) or is merely perceptual (Simple).
2. **Evaluation**: Compare your independent assessment with the [MODEL_JUDGMENT] to evaluate whether the model's judgment was accurate.

---

**2. INPUT SPECIFICATION**

You will receive three inputs:

1. **[IMAGE]**: The input image for analysis.
2. **[QUESTION]**: The question asked based on the image.
3. **[MODEL_JUDGMENT]**: The explanation from the model being evaluated.

---

**3. OUTPUT SPECIFICATION**

Your output **MUST** be a single JSON object parsed by `json.loads()`. It must contain:

1. `is_judgment_correct` (boolean): Is the model's judgment correct?
2. `requires_reasoning` (boolean): Does the question actually require reasoning?

---

**4. CORE ANALYSIS LOGIC**

1. **Step 1: Determine if the Question Truly Requires Reasoning**

   - **SIMPLE TASKS** (Perception) → `requires_reasoning`: **false**
     - *Definition*: Direct observation, recognition, or retrieval.
     - *Examples*: Visual Recognition, Counting, Simple OCR.
   - **COMPLEX TASKS** (Logic) → `requires_reasoning`: **true**
     - *Definition*: Inference, calculation, synthesis, or utilizing information from multiple parts of the image.
     - *Examples*: Deduction ("Why?"), Synthesis, Comparison.

2. **Step 2: Evaluate the Model's Judgment** (Value for `is_judgment_correct`)

   - Match your Step 1 result with the model's intent:
   - IF you say **true** AND model says "needed/complex" → **true**
   - IF you say **false** AND model says "no/simple" → **true**
   - Otherwise → **false**

---

**5. EXAMPLES**

**Example 1: The model correctly identifies a simple question.**
- [IMAGE]: [An image of a red fire hydrant]
- [QUESTION]: "What color is this fire hydrant?"
- [MODEL_JUDGMENT]: "No reasoning is needed because the answer can be obtained by directly observing the image."
- **Expected Output**:
  `{{ "is_judgment_correct":  true, "requires_reasoning":  false }}`

**Example 2: The model correctly identifies a complex question.**
- [IMAGE]: [A chart showing the price and features of Product A and Product B]
- [QUESTION]: "Which product offers better value for money?"
- [MODEL_JUDGMENT]: "Reasoning is required because it's necessary to compare prices and features to reach a conclusion."
- **Expected Output**:
  `{{ "is_judgment_correct":  true, "requires_reasoning":  true }}`

**Example 3: The model incorrectly classifies a complex question as simple.**
- [IMAGE]: [A line chart of a company's annual profit growth]
- [QUESTION]: "Based on this chart, what are the company's future prospects?"
- [MODEL_JUDGMENT]: "No reasoning is needed, this is a simple question."
- **Expected Output**:
  `{{ "is_judgment_correct":  false, "requires_reasoning":  true }}`

> ### Thinking Logic Reward Prompt
>
> **1. ROLE AND GOAL**
> You are a professor of logic and applied reasoning. Your task is to evaluate the **structural** and **deductive** soundness of a student's reasoning process. You must determine if the reasoning is valid from the initial setup (modeling based on Image/Question) to the final conclusion (execution).
>
> To do this, you will perform two rigorous checks:
> 1. **Structural Check**: Did the student correctly map the [IMAGE] and [QUESTION] into a valid logical or mathematical model?
> 2. **Deductive Check**: Is the step-by-step execution free of calculation errors, contradictions, or invalid logic?
>
> ---
>
> **2. INPUT SPECIFICATION**
> You will receive three inputs:
> 1. **[IMAGE]**: The visual context providing data or geometric information.
> 2. **[QUESTION]**: The specific problem asked based on the image.
> 3. **[THINKING]**: The student's step-by-step solution path to be evaluated.
>
> ---
>
> **3. OUTPUT SPECIFICATION**
> Your output **MUST** be a single JSON object parsed by `json.loads()`. It must contain:
> 1. `is_logically_sound` (boolean): Is the reasoning process completely valid (TRUE) or flawed (FALSE)?
>
> ---
>
> **4. CORE ANALYSIS LOGIC**
> 1. **Check 1: Structural Soundness (The Setup)**
>    - Does the chosen formula/logic correctly represent the principles described in the [QUESTION] and [IMAGE]?
>    - Are variables correctly mapped from the visual data? (e.g., correctly identifying "radius" vs "diameter" from the image).
>    - *Failure Rule*: If the starting formula is wrong, the logic is unsound, even if the math is perfect.
>
> 2. **Check 2: Deductive Soundness (The Execution)**
>    - Given the student's model (from Check 1), are the calculations correct?
>    - Is the algebraic manipulation valid within the [THINKING]?
>    - Are there any self-contradictions in the chain of thought?
>
> 3. **Final Judgment Rule (Strict AND Logic)**
>    - **TRUE** if AND ONLY IF both Structural and Deductive checks pass.
>    - **FALSE** if there is even a single, minor flaw in either structure or deduction.
>    - **Note**: You are not fact-checking empirical constants unless stated in the problem, but you ARE checking the logic of how they are used.
>
> ---
>
> **5. EXAMPLES**
> **Example 1: Structurally Unsound (Wrong Formula).**
> - [IMAGE]: [Diagram of a circle with radius labeled '5']
> - [QUESTION]: "Calculate the area of this circle."
> - [THINKING]: "Area = 2 * pi * r. So Area = 10pi." (Wrong formula used).
> - **Expected Output**: {{ `"is_logically_sound": false` }}
>
> **Example 2: Deductively Unsound (Calculation Error).**
> - [IMAGE]: [Image showing 2 apples costing $10]
> - [QUESTION]: "Calculate the cost of 1 apple (x)."
> - [THINKING]: "2x = 10. To find x, I divide by 2. x = 10 / 2. Therefore x = 4." (Math error).
> - **Expected Output**: {{ `"is_logically_sound": false` }}

**Example 3: Sound Reasoning.**
- [IMAGE]: [Logical diagram: A $\rightarrow$ B]
- [QUESTION]: "If A is true, what implies B?"
- [THINKING]: "By Modus Ponens, if A implies B and A is true, then B must be true."
- **Expected Output**: {{ `"is_logically_sound":  true` }}

---

## Thinking Consistency Reward Prompt

**1. ROLE AND GOAL**

You are a rigorous logic evaluator. Your task is to verify whether a given "Answer" is a direct and logically consistent result of the preceding "Thinking Process". You focus solely on the **consistency** between the reasoning steps and the final conclusion, disregarding external factual correctness.

To do this, you will perform two core tasks:

1. **Trace Reasoning**: Carefully read the steps in the [THINKING] to understand the derived logic.
2. **Verify Conclusion**: Determine if the [ANSWER] is the strict logical consequence of that process, without introducing new information or contradictions.

---

**2. INPUT SPECIFICATION**

You will receive two inputs:

1. **[THINKING]**: The step-by-step reasoning chain generated by the model.
2. **[ANSWER]**: The final conclusion or answer derived from the process.

---

**3. OUTPUT SPECIFICATION**

Your output **MUST** be a single JSON object parsed by `json.loads()`. It must contain:

1. `is_consistent` (boolean): Does the answer logically follow from the thinking process?

---

**4. CORE ANALYSIS LOGIC**

- **CRITERIA FOR "TRUE" (Consistent)**
  - The [ANSWER] is a direct summary or derivation of the final step in the [THINKING].
  - The logic flows smoothly from the reasoning to the conclusion without gaps.

- **CRITERIA FOR "FALSE" (Inconsistent)**
  - **Contradiction**: The Answer states $X$, but the Thinking Process argues for $Y$.
  - **Hallucination**: The Answer introduces new numbers, entities, or logic not mentioned in the Thinking Process.
  - **Disconnect**: The Answer is unrelated to the reasoning steps.

- **CRITICAL RULE**: Do **NOT** evaluate whether the Thinking Process is factually correct. Even if the reasoning is wrong (e.g., "1+1=3"), as long as the Answer matches that wrong reasoning ("Answer: 3"), it is **Consistent**.

---

**5. EXAMPLES**

**Example 1: Consistent (Even if factually wrong).**
- [THINKING]: "The sun is cold. Cold things are blue. Therefore the sun is blue."
- [ANSWER]: "The sun is blue."
- **Expected Output**: {{ `"is_consistent":  true` }}

**Example 2: Inconsistent (Contradiction).**
- [THINKING]: "Calculations show x = 5 and y = 10. So x + y = 15."
- [ANSWER]: "The answer is 20."
- **Expected Output**: {{ `"is_consistent":  false` }}

---

### Thinking Hallucination Reward Prompt

**1. ROLE AND GOAL**

You are a meticulous, multi-modal fact-checker. Your task is to assess if the provided [THINKING] is completely free of hallucinations by cross-referencing it against three verification sources: the Input Image, the Input Question, and Verifiable World Knowledge.

To do this, you will perform three specific grounding checks. A failure in **ANY** check means the content contains a hallucination.

---

**2. INPUT SPECIFICATION**

You will receive three inputs:
1. **[IMAGE]**: The visual context provided to the model.
2. **[QUESTION]**: The user's query or prompt constraints.
3. **[THINKING]**: The step-by-step reasoning or response generated by the model.

---

**3. OUTPUT SPECIFICATION**

Your output **MUST** be a single JSON object parsed by `json.loads()`. It must contain:
1. `is_hallucination_free` (boolean): Is the content free of any contradictions or fabrications? (TRUE = Clean, FALSE = Hallucinated).

---

**4. CORE ANALYSIS LOGIC**

1. **Check 1: Visual Grounding (The Eyes)**
   - Cross-reference every claim about visual elements against the [IMAGE].
   - Check for contradictions in: Objects (existence/count), Attributes (color/shape), Relationships (position/action), or Text in image (OCR).

2. **Check 2: Textual Grounding (The Instructions)**
   - Cross-reference claims against the [QUESTION].
   - Ensure the content respects stated constraints, data numbers, or specific conditions provided in the text.

3. **Check 3: World Knowledge (The Brain)**
   - For claims not verifiable by image/text, check against established facts (historical, scientific, geographical).
   - **Priority Exception**: If the [QUESTION] or [IMAGE] deliberately presents a hypothetical or counter-factual scenario (e.g., "Imagine the sky is green"), the provided context **overrides** world knowledge. The model should follow the context, not correct it.

4. **Final Judgment Rule**
   - **TRUE** (Pass): If ALL claims are supported by Image, Question, or Fact.
   - **FALSE** (Fail): If ANY single claim contradicts any source.

---

**5. EXAMPLES**

**Example 1: Visual Hallucination.**
- [IMAGE]: [A photo of two cats sleeping].
- [QUESTION]: "Describe the image."
- [THINKING]: "There are three dogs playing in the park."
- **Expected Output**: {{ `"is_hallucination_free":  false` }}

**Example 2: Knowledge Hallucination.**
- [IMAGE]: [Black image / Irrelevant content].
- [QUESTION]: "Who wrote Hamlet?"
- [THINKING]: "Hamlet was written by Charles Dickens."
- **Expected Output**: {{ `"is_hallucination_free":  false` }}

**Example 3: Context Override (Correct Handling).**
- [IMAGE]: [A fantasy landscape].

- **[QUESTION]:** "Assume that in this fantasy world, water boils at 10 degrees. At what temperature does water boil?"
- **[THINKING]:** "In this fantasy world, water boils at 10 degrees."
- **Expected Output**: {{ `"is_hallucination_free":  true` }}

## E  LLM USAGE

We utilized Large Language Models (LLMs), such as GPT-4, as a writing aid during the preparation of this manuscript. The use of LLMs was strictly limited to improving grammar, refining phrasing, and enhancing the overall clarity of the text. We did not use LLMs to generate any of the core content, ideas, or experimental results presented in this paper.

