# OpenReview forum: "SAIL-RL: Guiding MLLMs in When and How to Think via Dual-Reward Supervision"
_ICLR.cc/2026/Conference — ICLR 2026 Conference Desk Rejected Submission_

### Official Review · Reviewer_35oa · 2025-10-30

**Soundness:** 2
**Presentation:** 3
**Contribution:** 3
**Rating:** 4
**Confidence:** 4

**Summary:**

SAIL-RL proposes a novel reinforcement learning post-training framework. The motivation for this is to resolve two challenges: 1. Nowadays the reward is only designed based on the final answer without evaluation of reasoning steps 2. The models sometimes overthink on simple problems and simplifies the complex problems. The SAIL-RL consists of a dual-reward system:
- Thinking reward, which consists of Logical Coherence, Factual Grounding and Answer Consistency.
- Judging reward, justifies whether the problem is worth thinking.

The training strategy consists of two steps: LongCoT SFT and RL Finetuning. The evaluation results validates the models' superiority on multimodal reasoning benchmarks such as MathVista, LogicVista, and DynaMath.

**Strengths:**

1. The motivation of the SAIL-RL is good. The framework for RL post-training is novel, it teaches the model when to think and what to think.
2. The RL post-training framework can reduce the hallucination rate of SAIL-RL.
3. The evaluation set up is comprehensive, authors evaluate on both mathematical reasoning benchmarks like mathverse, mathvision, and also multimodal understanding benchmarks like MMMUval.
4. The structure of the paper is clear and easy to read.

**Weaknesses:**

1. In L296, **High computational cost**: the method use Gemini as reward model, as RL needs many roll out, this setup is computationally expensive and hard to reproduce.
2. **Stability**: Continuous request of Gemini is not stable for large-scale deployment.
3. **Potential Reward Hacking**: The solely usage of Gemini as reward model, the model may overfit to the pattern of Gemini, instead of truly improve reasoning abilities.
4. As illustrated in L433, 1.3% and 0.5% improvement is not significant gain, and **can not prove avoid overthinking is crucial**. I would also like to know is the experiment result only for one time inference or multi-times?
5. **Lack of the prompt and supplementary materials**: the paper lacks detailed prompt descriptions, which weakens its transparency, reproducibility, and interpretability. e.g. 1. the prompt for the judge? 2. the structure of the CoT thinking? 3. the format of final answer output? These significantly reduce the reproducibility of this paper.

**Questions:**

1. In line 291, the author claims that they have 100k data but in line 255 the author claims there is a high-quality dataset of 70K problems for RL training. What's the difference between those two datasets?
2. In L296, 100k data and many rollouts leads to 1000k+ requests, and consider the long chain-of-thought, so I want to know the cost for that. In the meantime, the prompt format? Could answer provides examples as I listed in weakness?
3. In Table2, The paper only reports the final SAIL-RL results. It would be helpful to include intermediate results (e.g., after one epoch of RL) to better illustrate the effect and learning progress of RL tuning.
4. L413, your method is blue? answer only? Is this a typo here?
5. In L433, is the experiment average after multiple inference or only once? I cannot see evidence for avoiding overthinking is crucial.

---

> ### Author Response · Authors · 2025-11-21
> **Response to Reviewer 35oa (Part-1/4)**
>
> > **Q1: The data size for RL training**.
>
> We sincerely appreciate the carefully question. We acknowledge that the mention of 100k was a clerical error. The correct size of the high-quality dataset used for RL training is indeed **70k samples** (comprising 50k STEM problems and 20k general domain problems). We have corrected this discrepancy in the revised manuscript.
>
> > **W1 & W2 & Part of Q2：Hard to reproduce due to High computational cost & Stability**
>
> Thank you for the concern regarding computational cost and stability. We fully agree that reproducibility and scalability are crucial for the community. Regarding your concerns, we wish to clarify the actual costs and provide our solutions for scalable deployment:
>
> **1. Clarification on Cost**
>
> The actual experimental cost is substantially lower than the initial estimation, based on two key facts:
> * **Reduced Request Scale:** As clarified in our response to the dataset size discrepancy, our experiments are based on the correct **70k samples** (not 100k). With 5 rollouts, this results in approximately **350k API requests**.
> * **Efficient Model Role:** As demonstrated by the prompt included in the **Appendix** of the revised manuscript, we utilize **Gemini-2.5-Pro** solely to output a JSON-formatted True/False signal, not to generate long chains of thought. Consequently, the cost is dominated by cheap input tokens (800tokens / request), while expensive output tokens are negligible (20tokens / request).
>
> Based on standard pricing ($1.25/1M Input, $10.00/1M Output), processing 350k requests consumes ~280M Input Tokens and ~7M Output Tokens.
> * **Single Epoch Cost:** Only **~$420**.
> * **Full Training Cost:** Even regenerating data for 3 full epochs costs only **~$1,260**.
> This demonstrates that our method is economically efficient and reproducible for most academic laboratories.
>
> **2. Stability and Large-Scale Deployment**
>
> To address concerns regarding API dependence and large-scale deployment, we provide two layers of safeguards:
> * **Enterprise Reliability:** During our experiments, we utilized the enterprise-grade Gemini API, which ensured stability under high concurrency and guarantees the reproducibility of our reported results.
> * **Local Deployment Solution (Distillation + Quantization):** To completely resolve cost and dependency issues for large-scale applications, we implemented a **local distillation strategy**. We fine-tuned a **Qwen-2.5-VL-32B** model using 500k reward samples generated by Gemini and applied **AWQ quantization** to reduce memory requirements.
>
> As shown below, the local model achieves **>95% agreement** with Gemini, proving that SAIL-RL can operate offline with high stability and low cost.
>
> | Reward Model | Reward Type | Agreement with Gemini |
> | :--- | :--- | :---: |
> | **Qwen-2.5-VL-32B-Distilled** | Judge Reward | **98.9%** |
> | | Thinking Reward | **97.2%** |
>
> In summary, we used Gemini in the paper primarily to establish a theoretical upper bound on performance and minimize noise. However, the distillation results confirm that the framework is fully capable of self-contained, stable local deployment.

---

> ### Author Response · Authors · 2025-11-21
> **Response to Reviewer 35oa (Part-2/4)**
>
> > **W3: Potential reward hacking due to only use Gemini**.
>
> Thank you for the helpful question. We address this concern by clarifying the structural role of the reward mechanism and providing experimental evidence using different reward models.
>
> **1. Clarification on the Reward mechanism**
>
> As detailed in the prompt provided in the appendix, Gemini is used strictly as a verifier that produces **discrete binary rewards (0/1)** according to a fixed prompt. Since SAIL-RL never observes the reward model's language style or intermediate reasoning traces, it is structurally impossible for it to imitate or overfit to Gemini’s behavior.
>
> **2. Robustness Across Different Reward Models**
>
> To show that SAIL-RL truly improves reasoning capability instead of overfit the output of  a specific model, we use SAIL-VL2-8B as the backbone and replace Gemini-2.5-Pro with **GPT-5** and **Qwen-2.5-VL-32B**.
>
> | Training | Reward Model (VLM-as-Judge) | **MathVision** | **LogicVista** | **MMMU** |
> | :--- | :--- | :---: | :---: | :---: |
> | **SFT** | - | 27.6 | 45.0 | 55.4 |
> | **RL** | **Gemini-2.5-Pro** | 49.6 | 63.8 | 66.1 |
> | **RL** | **GPT-5** | 49.7 | 63.5 | 66.4 |
> | **RL** | **Qwen-2.5-VL-32B** | 48.4 | 62.7 | 64.9 |
>
> As shown in the table above, SAIL-RL yields consistent improvements regardless of the reward model. For example, GPT-5 achieves 49.7 on MathVision, nearly identical to Gemini (49.6), while the local Qwen-32B also reaches 48.4, significantly outperforming the SFT baseline (27.6). These results support two distinct conclusions:
> **(1) Robustness against Overfitting:** The consistent improvements across diverse model families (Google, OpenAI, Alibaba) prove that SAIL-RL learns generalizable reasoning skills rather than overfitting to a specific judge's patterns.
> **(2) Viability of Local Judges:** While frontier models (GPT-5/Gemini-2.5-Pro) offer marginal gains due to higher precision of reward signal, the local Qwen-2.5-VL-32B model already provides sufficient supervision to achieve substantial improvements, demonstrating the framework's effectiveness even without proprietary APIs.
>
> > **Q3: The paper only reports the final SAIL-RL results in Table2. It would be helpful to include intermediate results (e.g., after one epoch of RL) to better illustrate the effect and learning progress of RL tuning**.
>
> Thank you for this valuable suggestion. We agree that showing the training dynamics provides valuable insight into the stability and efficiency of our RL process.
>
> We report the intermediate results based on the SAIL-VL2-8B at the end of each epoch during the RL training stage. As shown in the table below, the model exhibits a steady and consistent improvement trajectory.
>
> | Training Stage | **MathVision** | **LogicVista** | **MMMU** |
> | :--- | :---: | :---: | :---: |
> | **LongCoT SFT** (Epoch 0) | 39.7 | 58.2 | 63.0 |
> | **RL Epoch 1** | 45.2 | 61.4 | 64.5 |
> | **RL Epoch 2** | 48.5 | 63.0 | 65.6 |
> | **RL Epoch 3** (Final) | **49.6** | **63.8** | **66.1** |
>
> This progress demonstrates the robustness of our Dual Reward framework and the optimization process. A significant portion of the performance gain is realized in the first epoch (e.g., MathVision jumps from 39.7 to 45.2), indicating that the reward signal provides clear and effective guidance for rapid adaptation. Performance continues to improve steadily in subsequent epochs (2 and 3) without instability or degradation, confirming the robustness and smooth convergence of the learning process.
>
> > **Q4: L413 (Figure 5) is incorrect and should be flipped**.
>
> We sincerely thank the reviewer for the careful observation. Yes, this is indeed a typo; the legend labels in Figure 5 were inadvertently reversed. We have corrected this error in the revised manuscript to ensure that the method corresponds to the correct color code.

---

> ### Author Response · Authors · 2025-11-21
> **Response to Reviewer 35oa (Part-3/4)**
>
> > **W5 & Part of Q2: Lack of the prompt and supplementary materials, e.g. 1. the prompt for the judge? 2. the structure of the CoT thinking? 3. the format of final answer output? These significantly reduce the reproducibility of this paper**.
>
> Thank you for raising this important point regarding transparency and reproducibility. We have included all detailed prompts in the **Appendix** of the revised manuscript.
>
> To address the specific questions about the **Judge Prompt**, **CoT Structure**, and **Answer Format**, we provide the following clarification to resolve any ambiguity between the **Reward Model** and the **Policy Model**:
>
> **1. The Prompt for the Reward Model**
> To ensure deterministic reward derivation, our Reward Model Prompt is organized into five main components:
> * **Role and Goal:** Explicitly defines the role and specific tasks for the model.
> * **Input Specification:** Lists the specific inputs (`[IMAGE]`, `[QUESTION]`, `[MODEL_JUDGMENT]`) to ensure full context.
> * **Output Specification:** Enforces a strict JSON format to eliminate parsing ambiguity.
> * **Core Analysis Logic:** Establishes explicit evaluation criteria and step-by-step guidelines to determine the reward output based on the input.
> * **Examples:** Provides reference examples to guide the reward model's scoring.
>
> **2. Clarification on CoT and Output Format**
> We explicitly clarify the difference in output structure for the Reward Model versus the Policy Model:
>
> * **For the Reward Model (No CoT):** We clarify that the Reward Model **does not generate a Chain-of-Thought (CoT)** or free-form text. As defined in the **Output Specification** above, it strictly outputs a **JSON object** containing a binary signal (e.g., `{"is_judgment_correct": true}`). This design choice is crucial for ensuring the low computational cost and high stability of the reward signal.
> * **For the Policy Model (CoT):** The "structure of CoT thinking" refers to the policy model that is trained to generate outputs in a specific sequence: `[Judge statement] -> [Thinking Process] -> [Final Answer]`. Each component in this sequence is enclosed within a dedicated tag—`<judge></judge>`, `<think></think>`, and `\boxed{}` respectively—to allow for exact extraction and reward calculation.
>
> We hope that providing these exact specifications significantly enhances the reproducibility of our work. All these details are now explicitly documented in the revised paper.

---

> ### Author Response · Authors · 2025-11-21
> **Response to Reviewer 35oa (Part-4/4)**
>
> > **W4 & Q5: Table.5 can not prove avoiding overthinking is crucial. Are the results based on a single inference or averaged over multiple runs**.
>
> Thank you for the insightful concern regarding our when-to-think ablation studies. We address the questions below:
>
> **1. Inference Setting: Deterministic Evaluation**
>
> First, regarding the experiment reproducibility: We clarify that all reported results use **deterministic greedy decoding (temperature=0)**. Therefore, the performance differences do not stem from sampling randomness but reflect stable, intrinsic differences in model capability.
>
> **2. Analysis of 2B Model Results**
>
> We acknowledge that the average gain of **+0.6%** on the 2B model (Table 5) appears modest. However, this average does not fully reflect the Judge Reward’s contribution across different tasks: the intelligent allocation of reasoning . A task-level breakdown reveals its true impact:
> * **On perception-heavy tasks (General VQA):** These tasks require direct visual recognition , not multi-step reasoning. Forcing the model to "think" on such tasks could cause errors and hallucinations. The Judge Reward delivers a **significant gain of +1.2%**, directly validating its effectiveness in mitigating over-thinking on perception-heavy tasks.
> * **On reasoning-heavy tasks (Complex STEM):** These tasks require deep reasoning. The Judge Reward performs on par with Forced Thinking (**$\Delta \approx 0.1$**). This matches our expectation and proves that the Judge does not falsely suppress thinking on complex problems.
>
> **3. Validation Across Model Scales**
>
> To investigate how our method scales with model size, we conducted the same ablation on the **SAIL-VL2-8B** model. As shown in the table below, the benefits of the Judge mechanism are significantly amplified: the average improvement jumps from 0.6% (at 2B) to **3.1%** (at 8B), with a notable **+7.1%** boost on **MMVet**.
>
> This amplification occurs because the value of the Judge ("When to Think") is inherently coupled with the model's reasoning quality ("How well it Thinks"):
> * **At 2B (Limited Capacity):** The base reasoning capability is weaker. Even if the Judge correctly routes a hard problem to the reasoning path, the 2B model may still fail to solve it, capping the potential performance gain.
> * **At 8B (Strong Capability):** The model possesses strong reasoning skills. The Judge becomes highly effective here because it ensures this powerful capability is deployed *only* when needed (avoiding hallucinations on easy tasks) and *always* when necessary (solving hard tasks).
>
> In summary, these results confirm that the Judge mechanism is critical for unlocking the full potential of adaptive reasoning models.
>
> | Method | **MMMU** | **MMBench** | **MME** | **Hallucination** | **MMVet** | **Average** |
> | :--- | :---: | :---: | :---: | :---: | :---: | :---: |
> | **Forced Thinking** | 64.5 | 88.6 | 83.8 | 58.5 | 66.1 | 72.3 |
> | **Judge Reward (Ours)** | **66.1** | **90.4** | **86.0** | **61.5** | **73.2** | **75.4 (+3.1)** |

---

> ### Author Response · Authors · 2025-11-27
> **Gentle Reminder for Your Feedback**
>
> Dear Reviewer `35oa` :
>
> Thank you for your time and effort in reviewing our submission. We have carefully considered your comments and provided detailed responses. We look forward to your feedback.

---

> ### Comment · Reviewer_35oa · 2025-11-27
>
> Dear Authors,
>
> Thank you very much for your very thorough rebuttal. It effectivelly addresses my concerns. Good work. I will raise my scores accordingly.
>
> Best regards,
>
> Reviewer 35oa

---

> ### Author Response · Authors · 2025-11-27
> **Thank you for your feedback.**
>
> Dear Reviewer `35oa` :
>
> We are thrilled to hear that our rebuttal effectively addressed your concerns! We sincerely appreciate your recognition of our work and your decision to raise the score. To ensure the final version is as strong as possible, we remain fully available to answer further questions.

---

### Official Review · Reviewer_sUvu · 2025-10-31

**Soundness:** 3
**Presentation:** 3
**Contribution:** 3
**Rating:** 6
**Confidence:** 3

**Summary:**

This paper introduces SAIL-RL, a RL framework designed to improve the reasoning capabilities of MLLMs via a dual-reward design: thinking reward that evaluates the quality of the reasoning process, and judging reward that decides when to engage in deep reasoning or direct answering.

After applying this framework to the SAIL-VL2 model, and the resulting SAIL-VL2-Thinking shows state-of-the-art performance on multiple mathematical reasoning and general multimodal understanding benchmarks, outperforming comparable open-source models and competing with powerful closed-source models like GPT-4o.

**Strengths:**

1. To solve the lucky guess with flawed reasoning and over-thinking/simple-thinking in different questions, this paper design thinking reward and judge reward.
2. Propose a LongCoT dataset for SFT and extra dataset for RL training.
3. Equipping with SAIL-RL, SAIL-VL2 presents impressive performance on various benchmarks, significantly improving over its baseline and surpassing other leading open-source models.
4. The paper is well-written, logically structured, and easy to follow.

**Weaknesses:**

1. The components of the Thinking Reward (Logical Coherence, Factual Grounding, Answer Consistency) and the Judging Reward are all binary (0 or 1). A more continuous or granular scoring system might provide a richer and more accurate learning signal.
2. Are there specific types of problems or reasoning patterns where SAIL-RL struggles? For example, does it handle ambiguity or multi-step causal reasoning as effectively? An exploration of its boundaries would be interesting.
3. For thinking rewards, only average operation is used for integrate three sub-rewards. Did you try other options for the integration of different rewards?
4. Did you apply this RL framework in other open-source models beyond SAIL-VL2
5. Why the DAPO is used for RL training? Any other RL algorithms are tried.
6.It's better to give more details about the creation of ground truth labels based on task complexity.

**Questions:**

1. More continuous or granular scoring methods can be explored for reward.
2. How to define the ground truth labels based on task complexity?
3. Did the authors try different integration methods for different rewards
4. Proposed RL framework should be applied to some existing open-source MLLMs to see its perfmance improvement.
5. Any other RL optimization methods are explored?

---

> ### Author Response · Authors · 2025-11-21
> **Response to Reviewer sUvu (Part-1/3)**
>
> > **W1 & Q1: More continuous or granular scoring methods can be explored for reward**.
>
> Thank you for this valuable suggestion. We agree that in principle, a continuous scoring scheme (e.g., real-valued scores in [0,1]) could theoretically provide a richer learning signal.
>
> However, our empirical exploration suggests that **binary rewards currently yield superior results** for MLLM reasoning. During our early development, we directly compared continuous rewards (prompting the judge for a 0.0-1.0 score) against our binary approach on SAIL-VL2-8B.
>
> | Reward Type | **MathVision** | **LogicVista** | **MMMU** |
> | :--- | :---: | :---: | :---: |
> | **Continuous (0~1)** | 47.5 | 59.4 | 62.8 |
> | **Discrete (0/1) (Ours)** | **49.6** | **63.8** | **66.1** |
>
> As shown above, the discrete binary reward consistently outperforms the continuous variant. We attribute this counter-intuitive finding to two primary factors:
>
> **1. Ambiguity in LLM Judging (The Calibration Problem)**
>
> It is notoriously difficult for current LLMs to output calibrated, fine-grained continuous scores. Asking a judge to distinguish between a reasoning quality of "0.7" vs. "0.8" introduces significant subjective noise and inconsistency. In contrast, forcing a binary decision (e.g., *"Is this step logically coherent? Yes/No"*) yields a much sharper and more reproducible signal, reducing reward noise.
>
> **2. RL Optimization Stability (The Variance Problem)**
>
> RL training (specifically GRPO) is highly sensitive to the signal-to-noise ratio of the rewards. Noisy continuous scores increase the variance of the advantage estimation, destabilizing the optimization process. Binary rewards provide a high-discrimination signal that clearly separates "good" from "bad" behaviors, leading to more robust convergence.
>
> We view the design of finer-grained, low-variance continuous rewards as an important direction for future work, and our current empirical findings suggest that, under today’s LLM judges, binary rewards strike a practical balance between expressiveness and stability.
>
> > **W2: Are there specific types of problems or reasoning patterns where SAIL-RL struggles**.
>
> Thank you for this thoughtful suggestion. Exploring the boundaries of SAIL-RL is indeed crucial for understanding its current scope and future potential. We identify two primary reasoning patterns where SAIL-RL currently struggles:
>
> **1. Spatio-Temporal and Causal Reasoning (Dynamic Scenes)**
>
> * **Limitation:** The model is less effective on tasks that require reasoning over time, such as tracking event order in videos, understanding long-horizon physical interactions, or predicting future outcomes from a sequence of frames.
> * **Cause:** This stems from our data and reward design. The current Thinking and Judging Rewards are trained primarily on **static images**. Consequently, the "judge" lacks the temporal awareness necessary to evaluate causal chains in dynamic video inputs.
>
> **2. Interactive and Tool-Augmented Reasoning (External Actions)**
>
> * **Limitation:** The model struggles with tasks that require combining internal reasoning with external information retrieval or manipulation (e.g., using a calculator, checking a database, or precise OCR tool calling).
> * **Cause:** SAIL-RL focuses on optimizing the **internal** chain of thought. The current reward formulation does not yet cover multimodal "tool-use" steps, restricting its ability to handle problems where internal knowledge is insufficient.
>
> We will extend the reward framework from static images to spatio-temporal video reasoning, and evolve from purely internal thought to tool-augmented reasoning to broaden the model's real-world applicability.

---

> ### Author Response · Authors · 2025-11-21
> **Response to Reviewer sUvu (Part-2/3)**
>
> > **W3 & Q3: Did you try other options for the integration of thinking sub-rewards**.
>
> Thank you for this valuable question. To integrate the three sub-rewards—logical coherence ($r_{\text{logic}}$), factual grounding ($r_{\text{hallucination}}$), and answer consistency ($r_{\text{consistency}}$)—we adopted an equal-weight averaging strategy based on both theoretical principles and experimental verification.
>
> **1. Theoretical Motivation: Holistic Reasoning Quality**
>
> Our design principle is that a high-quality "thinking" process must be simultaneously logical, factually grounded, and consistent. We regard these three aspects as **equally important and complementary foundations** of reliable reasoning. If any one of them fails, the thinking process is fundamentally flawed. Therefore, assigning them equal weights provides the most balanced signal to guide the model.
>
> **2. Validation of Weighting Schemes**
>
> To validate this, we conducted an ablation study on SAIL-VL2-8B comparing our equal-weight design against biased weighting schemes. ($w_l$: Logic weight, $w_h$: Hallucination weight, $w_c$: Consistency weight)
> The results reveal a distinct **"see-saw" **effect: increasing the weight of a specific component often yields marginal gains on related tasks but leads to **degraded performance** on other benchmarks. These results confirm that equal-weight averaging yields the smoothest and most robust reward signal, justifying our final design choice.
>
> | Integration Weights ($w_l, w_h, w_c$) | **MathVision** | **LogicVista** | **MMMU** |
> | :--- | :---: | :---: | :---: |
> | **1/3, 1/3, 1/3** | **49.6** | **63.8** | **66.1** |
> | **1/2, 1/4, 1/4** | 50.0 (+0.4) | 63.3 (-0.5) | 65.8 (-0.3) |
> | **1/4, 1/2, 1/4** | 49.2 (-0.4) | 64.2 (+0.4) | 65.9 (-0.2) |
> | **1/4, 1/4, 1/2** | 49.3 (-0.3) | 63.5 (-0.3) | 66.5 (+0.4) |
>
> > **W4 & Q4: Proposed RL framework should be applied to some existing open-source MLLMs to see its performance improvement**.
>
> Thank you for this valuable suggestion. We fully agree that validating the SAIL-RL on diverse architectures is important for demonstrating its generality.
>
> To this end, we applied SAIL-RL to two open-source vision–language models of different scales: **Qwen-2.5-VL-3B** and **Qwen-2.5-VL-7B**. We compared the SFT baseline, a variant using only the answer reward (+ Answer Reward), and our full framework (+ Dual Reward).
>
> | Model | Training Type | **MathVision** | **LogicVista** | **MMMU** |
> | :--- | :--- | :---: | :---: | :---: |
> | **Qwen-2.5-VL-3B** | SFT | 18.1 | 36.0 | 48.1 |
> | | + Answer Reward | 25.2 | 41.2 | 49.2 |
> | | **+ Dual Reward (SAIL-RL)** | **27.3** | **42.9** | **51.7** |
> | **Qwen-2.5-VL-7B** | SFT | 25.4 | 47.9 | 58.1 |
> | | + Answer Reward | 27.1 | 51.7 | 61.2 |
> | | **+ Dual Reward (SAIL-RL)** | **30.2** | **53.4** | **63.1** |
>
> As shown in the table, our dual-reward framework consistently outperforms both the original SFT baselines and the answer-only variants across all benchmarks. For example, on Qwen-2.5-VL-7B, SAIL-RL improves the MathVision score to **30.2**, clearly surpassing both the SFT baseline (25.4) and the answer-only variant (27.1). Similar consistent gains are observed on MMMU (63.1 vs. 61.2) and on the smaller 3B model. These results provide strong evidence that the proposed design is not tied to a specific architecture or model size, but is a general and effective framework for enhancing complex reasoning abilities.

---

> ### Author Response · Authors · 2025-11-21
> **Response to Reviewer sUvu (Part-3/3)**
>
> > **W5 & Q5: Any other RL optimization methods are explored**.
>
> Thank you for this valuable question. It allows us to clarify our implementation choices.
>
> While our core contribution lies in the **Reward Design** (the novel Dual Reward framework) rather than the optimization algorithm itself, we aimed to select a method that is efficient, stable, and compatible with our reward structure. We chose DAPO for two primary reasons based on our preliminary exploration:
>
> **1. Superior Exploration**
> In the early stages of this project, we conducted a preliminary ablation comparing **DAPO** against **GRPO**. We observed that DAPO consistently yielded better training stability and a performance gain of approximately **1.5%** on average.
> This observation aligns with the findings in the original DAPO paper. Specifically, DAPO employs a distinct **CLIP range mechanism** (using **[0.2, 0.28]** compared to the standard **[0.2, 0.2]** in GRPO) and eliminates the explicit **KL penalty**. This design prevents the policy from being overly constrained by the reference model too early, thereby **enhancing the model's exploration capability** during the RL phase. This is particularly beneficial for discovering complex reasoning processes where diverse exploration is key.
>
> **2. Resource Efficiency**
> A significant practical advantage of DAPO is that it eliminates the need for loading an additional reference model during training. This substantially reduces GPU memory consumption, allowing us to allocate more resources to larger batch sizes and longer context windows, which are critical for training effective reasoning processes.
>
> Given the combination of enhanced exploration capabilities and resource efficiency, we adopted DAPO as the foundational optimization algorithm to best support our Dual Reward framework.
>
> > **W6 & Q2:  Give more details about the creation of ground truth labels based on task complexity**
>
> Thank you for the insightful question. We have added the judge reward prompt in the **Appendix** of the revised manuscript.
>
> We clarify that the criteria for determining task complexity (Simple vs. Complex) are **explicitly defined within the Judge Reward Prompt** itself. The Judge utilizes these definitions to generate the "ground truth" label for each input during the reward calculation process.  The details of Ground truth labels based on task complexity are shown below:
>
> **1. Complexity Criteria in Prompt**
>
> Specifically, the Judge Reward Prompt instructs the model to classify tasks based on the following logic (Section 4 of the Prompt):
>
> * **SIMPLE TASKS (Perception-centric):**
>     * **Definition:** Tasks where answers can be obtained via direct observation, recognition, or retrieval without multi-step reasoning.
>     * **Examples:** Visual Recognition, Counting, Simple OCR.
> * **COMPLEX TASKS (Logic-centric):**
>     * **Definition:** Tasks requiring inference, calculation, synthesis, or utilizing information from multiple parts of the image.
>     * **Examples:** Deduction, Synthesis, Comparison.
>
> **2. Validation of Prompt-Based Labels**
>
> To ensure these prompt-defined criteria align with human intent, we verified the labels generated by this Judge Prompt against human annotations on a random sample of 100 examples. We observed a **98% agreement rate**, confirming that the definitions in the prompt effectively capture the intended ground truth for task complexity.

---

> ### Author Response · Authors · 2025-11-27
> **Gentle Reminder for Your Feedback**
>
> Dear Reviewer `sUvu` :
>
> Thank you for your time and effort in reviewing our submission. We have carefully considered your comments and provided detailed responses. We look forward to your feedback.

---

> > ### Comment · Reviewer_sUvu · 2025-11-27
> >
> > Dear Authors,
> >
> > Thank you very much for your rebuttal. Your responses have addressed most of my concerns. I will keep my current positive rating.
> >
> > Best regards.

---

> ### Author Response · Authors · 2025-11-27
> **Thank you for your feedback**
>
> Dear Reviewer `sUvu` :
>
> We sincerely thank you for keeping the positive rating. We would welcome the opportunity to address any remaining points to fully resolve your concerns and improve our paper.

---

### Official Review · Reviewer_aBKU · 2025-11-01

**Soundness:** 3
**Presentation:** 2
**Contribution:** 2
**Rating:** 4
**Confidence:** 3

**Summary:**

Existing RL approaches on MLLMs are limited by outcome-only supervisionand uniform thinking strategies.
This paper introduces a dual reward system that explicitly supervises both the quality of reasoning and the adaptivity of thinking strategies.

Experiments on SAIL-VL2 show that SAIL-RL improves reasoning and multimodal understanding benchmarks at both 4B and 8B scales, achieves state-of-the-art results among models of comparable size.

**Strengths:**

- The paper proposes a dual reward system to address two key issues: the lack of rewards for reasoning trajectories and the insufficient adaptivity of thinking strategies.
- The authors perform extensive experiments to validate the effectiveness of the proposed method.

**Weaknesses:**

- The related work section should include a more detailed discussion of existing RL methods for MLLMs, and explicitly clarify how the proposed approach differs from prior works.
- Table 1 omits the performance results of the base model (SAIL-VL). Also, please clarify whether SAIL-VL-Instruct refers to the model fine-tuned with the proposed LongCoT dataset or the base model itself. The paper should include the results of the base model, the SFT model (after stage 1), and the final model, to better illustrate the performance improvement achieved at each training stage.
- The legend in Figure 5 is incorrect and should be flipped
- The paper lacks sufficient details on the prompts used to instruct Gemini as a reward judge. It is also unclear how the reward scores are derived from Gemini’s textual outputs.
- The paper introduces a cascading product formulation for combining the judge, think, and answer rewards instead of an additive combination. It would be good if the paper could provide reference or ablation studies validating this design choice.

**Questions:**

see weakness

---

> ### Author Response · Authors · 2025-11-21
> **Response to Reviewer aBKU (Part-1/3)**
>
> > **W1: The related work section should include a more detailed discussion of existing RL methods for MLLMs, and explicitly clarify how the proposed approach differs from prior works**.
>
> Thank you for the detailed feedback. We have substantially revised the Related Work section in the updated manuscript to better demonstrate our contributions within the existing RL for MLLMs. Here, we summarize the two key differences we emphasized to clarify our novelty:
>
> **1. Distinction from traditional RL for MLLMs (Outcome vs. Process)**
> We clarify that most existing RL methods for MLLMs treat the model as a "black box" that is optimized solely based on the final response quality. In contrast, SAIL-RL explicitly models the reasoning process. We discuss how our approach advances beyond standard alignment by using a **dual-reward system** to optimize both the decision to reason and the quality of the reasoning steps themselves.
>
> **2. Uniqueness of the Reward Mechanism (Dynamic vs. Static)**
> We explicitly contrast our specific reward designs with other recent approaches to highlight our novelty:
> * **Regarding "When to Think" (vs. Length-based Rewards):** Unlike methods that use simple length penalties or rewards to control verbosity, our R-judge learns a sophisticated, context-dependent policy. We argue that static length constraints are insufficient; SAIL-RL teaches the model to recognize *specific contexts* where reasoning is beneficial, achieving a dynamic allocation of compute that simple length rewards cannot match.
> * **Regarding "How to Think" (vs. Standard Process Rewards):** We clarify how our method differs from standard process supervision. Instead of treating all steps equally, our design coordinates the R-judge (the gatekeeper) with fine-grained process rewards. This ensures the model is not just generating correct steps, but is doing so only when the judge deems it necessary, creating a more efficient reasoning pipeline than previous works.
>
> > **W2: The paper should include the results of the base model, the SFT model (after stage 1), and the final model, to better illustrate the performance improvement achieved at each training stage**.
>
> Thank you for this helpful suggestion. In the revised version, we have clarified that **SAIL-VL2-8B-Instruct** refers to the standard instruction-tuned model (Basic SFT), **SAIL-VL2-8B-LongCoT** refers to the model fine-tuned with our LongCoT dataset, and **SAIL-VL2-8B-Thinking** refers to the final model optimized via our dual-reward RL.
>
> We have updated Table 1 and Table 2 in the revised manuscript to explicitly report the performance of these three variants. For the reviewer's convenience, we present the results on several representative benchmarks below to illustrate the improvement at each stage:
>
> | Model | MathVision | LogicVista | MMMU |
> | :--- | :---: | :---: | :---: |
> | **SAIL-VL2-8B-Instruct** (Basic SFT) | 27.6 | 45.0 | 55.4 |
> | **SAIL-VL2-8B-LongCoT** (LongCoT SFT) | 39.7 | 58.2 | 63.0 |
> | **SAIL-VL2-8B-Thinking** (RL) | **49.6** | **63.8** | **66.1** |
>
> From these results, we observe that **SAIL-VL2-8B-LongCoT** achieves noticeable gains over the standard **SAIL-VL2-8B-Instruct** (e.g., +12.1% on MathVision), as it teaches the model to produce structured long-form reasoning. However, we also find that the LongCoT model tends to generate overly long or sometimes unreliable reasoning chains, especially in challenging scenarios. This behavior confirms that supervised fine-tuning on LongCoT alone is not sufficient to obtain robust and efficient reasoning. Our RL stage (**SAIL-VL2-8B-Thinking**) is therefore designed to further refine both **“when to think”** (optimizing efficiency) and **“how to think”** (enhancing effectiveness), aligning the reasoning process with overall task success. We believe the updated tables and explanations make this multi-stage improvement trajectory and the role of RL much more transparent.
>
> > **W3: The legend in Figure 5 is incorrect and should be flipped**.
>
> We sincerely thank you for carefully checking the figure. We have corrected the error in **Figure 5** in the revised manuscript to ensure accurate illustration.

---

> ### Author Response · Authors · 2025-11-21
> **Response to Reviewer aBKU (Part-2/3)**
>
> > **W4: The paper lacks sufficient details on the prompts used to instruct Gemini as a reward judge. It is also unclear how the reward scores are derived from Gemini’s textual outputs**.
>
> Thank you for raising this important point about the design of the reward prompt. To ensure full reproducibility, we have included the reward prompt in the **Appendix** of the revised manuscript.
>
> To clearly answer the questions, we present the exact content of our **Judge Reward Prompt** below, followed by the derivation logic.
>
> **1. Prompt Structure Design**
> The prompt is organized into five main components to ensure deterministic reward derivation:
> * **Role and Goal:** Explicitly defines the role and specific tasks for the model.
>
> * **Input Specification:** Lists the specific inputs (`[IMAGE]`, `[QUESTION]`, `[MODEL_JUDGMENT]`) to ensure full context.
> * **Output Specification:** Enforces a strict JSON format to eliminate parsing ambiguity.
> * **Core Analysis Logic:** Establishes explicit evaluation criteria and step-by-step guidelines to determine the reward output based on the input.
> * **Examples:** Provides reference examples to guide the reward model's scoring.
>
> **2. Derivation of Reward Scores**
> Based on the prompt structure above, our reward derivation process follows a strict three-step pipeline:
> 1. **JSON Format Enforcement:** As defined in the **Output Specification**, the judge is strictly constrained to output a single JSON object. This prevents free-form text generation that causes parsing failures.
> 2. **Deterministic Parsing:** We parse the valid JSON object using `json.loads()`.
> 3. **Boolean-to-Scalar Mapping:** We extract the `is_judgment_correct` boolean field and map it directly to a scalar reward (e.g., $1$ for `True`, $0$ for `False`). This provides a sharp, noise-free learning target for the policy.
>
> ------
> **Judge Reward Prompt**
>
> **1. Role and Goal**
> You are a top-tier AI logic analyst. Your task is to generate a ground truth assessment of task complexity and evaluate whether a "model to be evaluated" has correctly judged if a question requires "reasoning".
>
> To do this, you will perform two core tasks:
> 1. **Independent Assessment**: Analyze the `[QUESTION]` and `[IMAGE]` yourself to determine if the question truly requires reasoning (Complex) or is merely perceptual (Simple).
> 2. **Evaluation**: Compare your independent assessment with the `[MODEL_JUDGMENT]` to evaluate whether the model's judgment was accurate.
>
>
> **2. Input Specification**
> You will receive three inputs:
> 1. `[IMAGE]`: The input image for analysis.
> 2. `[QUESTION]`: The question asked based on the image.
> 3. `[MODEL_JUDGMENT]`: The explanation from the model being evaluated.
>
>
> **3. Output Specification**
> Your output **MUST** be a single JSON object parsed by `json.loads()`. It must contain:
> 1. `is_judgment_correct` (boolean): Is the model's judgment correct?
> 2. `requires_reasoning` (boolean): Does the question actually require reasoning?
>
>
> **4. Core Analysis Logic**
>
> **Step 1: Determine if the Question Truly Requires Reasoning**
> * **SIMPLE TASKS (Perception)** → `requires_reasoning`: **false**
>     * *Definition*: Direct observation, recognition, or retrieval.
>     * *Examples*: Visual Recognition, Counting, Simple OCR.
> * **COMPLEX TASKS (Logic)** → `requires_reasoning`: **true**
>     * *Definition*: Inference, calculation, synthesis, or utilizing information from multiple parts of the image.
>     * *Examples*: Deduction ("Why?"), Synthesis, Comparison.
>
> **Step 2: Evaluate the Model's Judgment** (Value for `is_judgment_correct`)
> * Match your Step 1 result with the model's intent:
> * IF you say **true** AND model says "needed/complex" → **true**
> * IF you say **false** AND model says "no/simple" → **true**
> * Otherwise → **false**
>
>
> **5. Examples**
>
> **Example 1: The model correctly identifies a simple question.**
> * `[IMAGE]`: [An image of a red fire hydrant]
> * `[QUESTION]`: "What color is this fire hydrant?"
> * `[MODEL_JUDGMENT]`: "No reasoning is needed because the answer can be obtained by directly observing the image."
> * **Expected Output**: `{"is_judgment_correct": true, "requires_reasoning": false}`
>
> **Example 2: The model correctly identifies a complex question.**
> * `[IMAGE]`: [A chart showing the price and features of Product A and Product B]
> * `[QUESTION]`: "Which product offers better value for money?"
> * `[MODEL_JUDGMENT]`: "Reasoning is required because it's necessary to compare prices and features to reach a conclusion."
> * **Expected Output**: `{"is_judgment_correct": true, "requires_reasoning": true}`

---

> ### Author Response · Authors · 2025-11-21
> **Response to Reviewer aBKU (Part-3/3)**
>
> > **W5: It would be good to provide ablation studies validating this design choice of cascading product formulation for combining the judge.**
>
> Thank you for this insightful suggestion, which touches upon the core philosophy of our reward mechanism design. Our main motivation is that **a successful reasoning process should form a correct and coherent chain**: a reliable decision of whether to think (judge), a consistent and useful reasoning process (think), and an accurate final prediction (answer).
>
> **1. Theoretical Motivation**
>
> The **Cascading Product** formulation ($R_{total} = R_{judge} \times R_{think} \times R_{answer}$) perfectly implements the philosophy described above.
> * **Cascading Product:** This design enforces **conditional dependency**. It ensures that the total reward is maximized *only* when every link in the chain is valid. If any step fails (e.g., the model decides to think when unnecessary, or generates correct reasoning but a wrong answer), the multiplicative nature suppresses the total reward, strictly penalizing error propagation.
> * **Additive Combination:** In contrast, an additive approach ($R_{total} = (R_{judge} + R_{think} + R_{answer}$)/3) treats these components independently. This creates room for **reward hacking**: the model may compensate for a poor judge decision or noisy reasoning trace by only optimizing the final answer reward. This weakens the pressure to improve the full reasoning chain and undermines our goal of shaping both “when to think” and “how to think” consistently.
>
> **2. Experimental Validation**
>
> To verify this hypothesis, we conducted an ablation study comparing the **Cascading Product** against an **Additive Combination** ($R_j + R_t + R_a$) on the SAIL-RL-8B.
>
> | Reward Formulation | **MathVision** | **LogicVista** | **MMMU** |
> | :--- | :---: | :---: | :---: |
> | **Additive Combination** | 46.1 | 60.7 | 63.8 |
> | **Cascading Product (Ours)** | **49.6** | **63.8** | **66.1** |
>
> The results above demonstrate that the **Cascading Product** consistently outperforms the Additive Combination across all benchmarks. The gap is particularly notable on **MathVision** (+3.5%), a benchmark requiring rigorous multi-step reasoning. This performance advantage confirms that coupling the rewards multiplicatively encourages the model to treat the reasoning process as a holistic entity, leading to more robust performance than merely accumulating independent rewards.

---

> ### Author Response · Authors · 2025-11-27
> **Gentle Reminder for Your Feedback**
>
> Dear Reviewer `aBKU `:
>
> Thank you for your time and effort in reviewing our submission. We have carefully considered your comments and provided detailed responses. We look forward to your feedback.

---

> > ### Comment · Reviewer_aBKU · 2025-11-27
> >
> > Thank you for your detailed response. It resolved most of my concerns, and I would like to increase my score.

---

> ### Author Response · Authors · 2025-11-27
> **Thank you for your feedback**
>
> Dear reviewer  `aBKU`,
>
> Thank you for the detailed response and the decision to increase the score. It is encouraging to know that most of your concerns have been resolved. We are happy to engage in further discussion to improve the quality of our work.

---

### Official Review · Reviewer_MRiy · 2025-11-03

**Soundness:** 2
**Presentation:** 2
**Contribution:** 2
**Rating:** 4
**Confidence:** 4

**Summary:**

The paper proposes an RL post-training framework for multimodal LLMs that teaches both *when to think* and *what to think*. The method adds two learning signals on top of the RLVR answer and format rewards: a Thinking Reward that scores logic, factual grounding, and answer consistency, and a Judging Reward that determines whether to trigger detailed reasoning or provide an answer directly. Gemini-2.5-Pro serves as the reward judge during RL. Results report gains on OpenCompass reasoning and general V+L benchmarks and show adaptive “thinking trigger” rates by task type.

**Strengths:**

* outcome-only RL can reward lucky guesses and uniform “always think” policies. The dual rewards target both reasoning quality and adaptivity.
* logic, grounding, and consistency are explicitly checked in the Thinking Reward. The total reward uses a multiplicative “AND-gate” to reduce reward hacking.
* Figure 4 reports low trigger rates and high rates on math-heavy ones, suggesting resource allocation by task type.
* The 8B “Thinking” model improves over its “Instruct” base and is strong among open-source models on both reasoning and general V+L benchmarks.

**Weaknesses:**

1. The Thinking Trigger analysis (Figure 4: “Evaluation results on thinking trigger”) only shows the adaptive allocation of reasoning resources after training. It will be good also to compare at least: a) the original model without training, b) GRPO with only R-answer and no judge in the output, and c) a variant that includes the judge output but removes the *when to think* reward. These baselines would show whether SAIL-RL truly teaches a sharper judge for *when to think* and by how much.
2. Beyond Figure 4, the *when to think* analysis is thin. For example, compared to a model trained without the *when to think* reward, by how much does total token usage drop overall?
3. Related work is weak, which makes it hard to assess novelty and the paper’s uniqueness relative to recent work. Literature on System 1 and System 2 thinking is relevant here, as the judge essentially serves as a router that chooses between the two. Additionally, the main design novelty lies in rewards; therefore, the RL section should discuss recent designs, such as work that adds length-relevant rewards (for your *when to think* design) and others that incorporate fine-grained LLM-Judge-based process rewards for the thinking process (for your *what to think* design), for a more thorough examination of GRPO-style RLVR.

**Questions:**

1. I am curious whether you observe some cases where SAIL-VL2-Thinking judges that thinking is needed, but during the thinking process, it realizes the initial judgment was wrong and the problem can be solved quickly. The reverse may also occur. Intuitively, this is possible. Humans also misjudge difficulty at first. Unlike your current output format, humans can adjust the decision dynamically while thinking. Does the model encounter this, and if so, what happens as a result?
2. The legend in Figure 5 looks reversed.
3. One point of confusion: the training stage uses Gemini-2.5-Pro as the VLM-Judge, while evaluation uses GPT-4o-Mini as the judge. I am not sure GPT-4o-Mini is strong enough as an evaluator, given that some baselines include stronger models like GPT-4o. Why use a weaker model as the LLM judge for evaluation? Since you can use Gemini-2.5-Pro, you should also be able to use a model of similar strength, for example, GPT-5 or at least GPT-5-mini.

---

> ### Author Response · Authors · 2025-11-21
> **Response to Reviewer MRiy (Part-1/3)**
>
> > **W1 & W2: The impact of the R-judge (when to think ) reward on the Thinking Trigger needs to be analyzed through a more comprehensive ablation study.**
>
> Thank you for the constructive suggestion. To address this concern, we conducted additional experiments to systematically analyze the “when to think” judge reward in SAIL-RL by comparing it against three baselines that match the reviewer’s proposals.
>
> -   **“never-think”:** The original SFT model without any RL training, which never triggers thinking.
> -   **“always-think”:** A variant where we apply DAPO only on the final answer and modify the system prompt to remove the judge token, forcing the model to think on all inputs.
> -   **“judge-without-reward”:** A variant where the model still produces a judge output, but the when-to-think reward is removed, so the model is not explicitly optimized for deciding when to think.
>
> We evaluate these variants and SAIL-RL on two complementary benchmarks using the same base model SAIL-VL2-8B-Instruct. **MathVision** focuses on reasoning-centric tasks, where thinking is expected to be beneficial. **OCRBench** focuses on perception-heavy tasks, where many examples can be solved without long chains of thought.
>
> As shown in Table below,  we demonstrate the critical role of our R-judge reward in three aspects:
> - The R-judge reward **teaches the model an adaptive reasoning policy**. We see it successfully activating reasoning for nearly all complex MathVision problems (99.8% trigger rate) while correctly suppressing it for most simpler OCRBench tasks (7.5% trigger rate).
> - The R-judge reward **leads to superior reasoning performance**. By learning when to think, the model avoids the inefficiency of naive "Always-Think" approaches. On OCRBench, this adaptive reasoning makes SAIL-RL both more accurate (91.3% vs. 88.7%) and nearly 4x more token-efficient.
> - The R-judge reward **is proven to be the essential ingredient**. The "Judge-without-reward" baseline, which has the mechanism to judge but lacks the explicit reward signal, fails to learn an effective policy and performs poorly. This directly confirms that our R-judge reward is the key driver for teaching the model intelligent resource allocation, not merely the presence of a judge token.
>
> | Model Strategy | MathVision | | | OCRBench | | |
> | :--- | :---: | :---: | :---: | :---: | :---: | :---: |
> | | **Thinking trigger** | **Accuracy** | **Token Usage** | **Thinking trigger** | **Accuracy** | **Token Usage** |
> | **Never-think** | 0% | 27.6 | 1.0x (baseline) | 0% | 90.5 | 1.0x (baseline) |
> | **Always-think** | 100% | 48.7 | 5.4x | 100% | 88.7 | 4.7x |
> | **Judge-without-reward** | 90.4% | 47.5 | **4.6x** | 47.6% | 89.8 | 2.9x |
> | **SAIL-RL** (Ours) | **99.8%** | **49.4** | 5.1x | **7.5%** | **91.3** | **1.2x** |
>
> > **W3: The paper's contribution could be further clarified by expanding the related work to include "System 1/2 thinking" theories and recent RL reward designs relevant to "when to think" and "how to think"**.
>
> Thank you for the constructive feedback on the related work. Following this valuable suggestion, we have revised and expanded this section in the updated manuscript. The key improvements are as follows:
>
> *   **Connecting to System 1 / System 2 Thinking:** We have incorporated a discussion on dual-process theory, framing our judge mechanism as a learned router that chooses between fast (System 1) and slow (System 2) thinking, and connecting it to recent work in this area.
>
> *   **Situating Our Reward Design within Recent RL Advances:** We have expanded our discussion of reward designs to better highlight the novelty of our approach. Specifically, we now:
>     *   Contrast our R-judge reward (for "when to think") with methods that use simpler, less flexible length-relevant rewards.
>     *   Clarify how our R-think reward (for "how to think") differ from recent work on step-by-step reward which is hard to label.

---

> ### Author Response · Authors · 2025-11-21
> **Response to Reviewer MRiy (Part-2/3)**
>
> > **Q1: How does the model handle suboptimal decisions from the judge? Specifically, is there a mechanism to recover from false negatives (i.e., the judge failing to trigger reasoning when it is needed), or to mitigate the inefficiency of false positives (i.e., triggering unnecessary reasoning)**
>
> Thank you  for this insightful question. Our framework aims to make the initial judgment as accurate as possible, rather than to dynamically correct it in the following thinking process. To ensure its accuracy, our training process, guided by the R-judge reward, strictly penalizes two distinct types of errors:
>
> *   **1. Incorrect Judgment (A Cognitive Error):** The judge's decision (to think or not) is compared against a ground-truth label. If this initial judgment is wrong, R-judge is immediately set to zero. This rule punishes a failure in correctly assessing the problem's difficulty.
>
> *   **2. Inconsistent Action (An Execution Error):** Even if the judgment is correct, the model's subsequent actions must align with its decision. If it decides to think but provides no reasoning, or decides not to think but generates unnecessary steps, R-judge is also set to zero. This rule punishes a failure to faithfully execute its own plan and prevents the model from "gaming" the system.
>
> As the total reward is a **cascade product**, The penalty for committing either of these errors is **severe** :
> $$ \text{Reward} = 0.9 \cdot R_{\text{judge}}  \cdot R_{\text{think}} \cdot R_{\text{answer}} + 0.1 \cdot R_{\text{format}} $$
>
> Under this design, once R-judge becomes zero, the main part of the reward collapses to zero, no matter how good the reasoning or final answer is. This creates a strong training signal that heavily penalizes any misstep in the initial judgment or its execution. However, we fully agree, that enabling the model to revise its decision is an interesting and important direction, and we leave it as a promising future work.
>
> > **Q2: The legend in Figure 5 looks reversed.**
>
> Thank you for carefully checking the figures and pointing this out. We acknowledge that the legend in **Figure 5** was reversed in the original submission. We have corrected this error and updated the figure accordingly in the revised manuscript.

---

> ### Author Response · Authors · 2025-11-21
> **Response to Reviewer MRiy (Part-3/3)**
>
> > **Q3: Why choose a weaker model (GPT-4o-Mini) as the evaluation judge, given that a much stronger model (Gemini-2.5-Pro) is used for training**
>
> Thank you for raising the important question. We understand the concern that (1) GPT-4o-Mini may appear insufficiently strong as an evaluator compared to some baselines, and (2) our method should not implicitly depend on the specific judge used during training.
>
> **1. Evaluation Model: Why GPT-4o-Mini is Sufficient**
>
> Our choice of GPT-4o-Mini as the evaluation model is based on three key reasons:
> *   **Following Community Standards:** We use GPT-4o-Mini primarily to adhere to the default configuration of the widely-used **`vlmevalkit`** framework, ensuring a fair and consistent comparison with other methods.
> *   **Nature of the Task:** The benchmarks (e.g., MathVision,) have **objective, ground-truth answers**. The evaluator's role is not open-ended reasoning, but a more constrained task: parsing the model's final output and checking if it matches the correct answer. GPT-4o-Mini is more than capable of performing this structured matching task reliably.
> *   **Empirical Verification:** Most importantly, we conducted additional experiments to verify this choice. We re-evaluated our model using several **stronger evaluators** (GPT-5 and Gemini-2.5-Pro). The results, summarized below, showed that the final scores **fluctuated by less than 0.2 points**. This confirms that our reported gains are stable and not dependent on a specific evaluator; GPT-4o-Mini is indeed a reliable choice for this task.
>
> | Evaluator      | MathVision | LogicVista | MMMU |
> | :------------- | :--------: | :--------: | :--: |
> | GPT-4o-Mini    |    49.4    |    63.8    | 66.1 |
> | GPT-5          |    49.4    |    63.6    | 66.0 |
> | Gemini-2.5-Pro |    49.6    |    63.8    | 66.1 |
>
> **2. Reward Model: Why Gemini-2.5-Pro is Necessary**
>
> In contrast, the RL training stage need an accurate and robust reward signal, making a stronger model like Gemini-2.5-Pro essential:
>
> *   **Complexity of the Task:** The training judge must **interpret multi-step reasoning traces, identify subtle logical errors, and jointly assess the final answer and the intermediate thinking process** to provide informative R-answer and`R-think rewards.
> *   **Necessity of High-Quality Rewards:** GRPO-style RL is **critically dependent on the quality of the reward signal**. We employ Gemini-2.5-Pro to generate these high-fidelity rewards, ensuring the accuracy and stability needed to effectively teach the model both what to think and how to think.
>
> The additional experiments on SAIL-VL2-8B-Instruct show that our method is robust to the choice of reward model. Meanwhile, using a stronger reward model is important to provide high-quality reward signal for our RL training.
>
> | Training      | Reward Model    | MathVision | LogicVista | MMMU |
> | :------------ | :-------------- | :--------: | :--------: | :--: |
> | SFT           | -               |    27.6    |    45.0    | 55.4 |
> | RL        | Gemini-2.5-Pro |  49.6  |  63.8  | 66.1 |
> | RL            | GPT-5           |    49.7    |    63.5    | 66.4 |
> | RL            | Qwen2.5-VL-32B  |    48.4    |    62.7    | 64.9 |

---

> ### Author Response · Authors · 2025-11-27
> **Gentle Reminder for Your Feedback**
>
> Dear Reviewer `MRiy`:
>
> Thank you for your time and effort in reviewing our submission. We have carefully considered your comments and provided detailed responses. We are looking forward to your feedback.

---

### Note · Program_Chairs · 2026-01-17
**Submission Desk Rejected by Program Chairs**

The following references in this submission do not refer to real documents and/or have major errors in bibliographic information:

 Zhen Fang, Jiacheng Li, Lichan Zhang, Pan Lu, Bodhisattwa Prasad Majumder, Tony Xia, Rami
Al-Rfou, and Swaroop Mishra. Wemath: A well-crafted dataset for mathematical reasoning with
weak supervision. arXiv preprint arXiv:2405.19228, 2024.